Residency, movement patterns, behavior and demographics of reef manta rays in Komodo National Park

http://orcid.org/0000-0002-8797-4451 Germanov Elitza S. 1 2 3 elitza@marinemegafauna.org
http://orcid.org/0000-0002-9375-5175 Pierce Simon J. 1
http://orcid.org/0000-0003-1886-5355 Marshall Andrea D. 1
Hendrawan I. Gede 4
Kefi Ande 5
Bejder Lars 2 3 6
http://orcid.org/0000-0003-2536-8656 Loneragan Neil 2 3 7
1 Marine Megafauna Foundation , West Palm Beach, Florida , United States of America
2 Centre for Sustainable Aquatic Ecosystems, Harry Butler Institute, Murdoch University , Perth, Western Australia , Australia
3 Environmental and Conservation Sciences, Murdoch University , Perth, Western Australia , Australia
4 Faculty of Marine Sciences and Fisheries, Universitas Udayana , Denpassar, Bali , Indonesia
5 Komodo National Park , Labuan Bajo, Flores , Indonesia
6 Marine Mammal Research Program, Hawaii Institute of Marine Biology, University of Hawaii at Manoa , Honolulu, Hawaii , United States
7 Faculty of Fisheries and Marine Science, Bogor Institute of Agriculture , Bogor, West Java , Indonesia
Costello Mark
Electronic publication date: 2022 May 16
Publication date: 2022
Volume: 10
Electronic Location ID: e13302
Received 2021 Oct 4; Accepted 2022 Mar 29
Copyright: © 2022 Germanov et al.
Copyright year: 2022
Copyright holder: Germanov et al.
License: This is an open access article distributed under the terms of the Creative Commons Attribution License, which permits unrestricted use, distribution, reproduction and adaptation in any medium and for any purpose provided that it is properly attributed. For attribution, the original author(s), title, publication source (PeerJ) and either DOI or URL of the article must be cited.
License URL: https://creativecommons.org/licenses/by/4.0/

Keywords: Elasmobranch, Photo-ID, Population structure, Movement, Site use, Tourism, Fisheries, Citizen science, Coral Triangle Region, MPA

Funding: Australian Postgraduate Award & Murdoch International Top Up 32608315 Ocean Park Conservation Foundation FH04_1516 Foundation Fortuna Idea Wild PADI Foundation 14668 Mantahari Oceancare Fish-Are-Friends SeaMorgens While no specific funding was received for this research, Elitza S. Germanov is a recipient of an Australian Postgraduate Award & Murdoch International Top Up (32608315) and received doctoral project funding from Ocean Park Conservation Foundation (FH04_1516) for fieldwork and laboratory work; Foundation Fortuna for fieldwork; Idea Wild for equipment; PADI Foundation (14668) for fieldwork; Mantahari Oceancare, Fish-Are-Friends, SeaMorgens, R. Horner and private donors, for general manta ray research, education, and conservation programs. The funders had no role in study design, data collection and analysis, decision to publish, or preparation of the manuscript.

==============================
Background

The reef manta ray (Mobula alfredi) is a globally threatened species and an iconic tourist attraction for visitors to Indonesia’s Komodo National Park (NP). In 2013, manta ray fishing was banned in Komodo NP and its surroundings, preceding the nationwide manta ray protection in 2014. Over a decade ago, a previous acoustic telemetry study demonstrated that reef manta rays had high fidelity to sites within the park, while more recent photo-identification data indicated that some individuals move up to 450 km elsewhere. Characterization of manta ray demographics, behavior, and a focused assessment on site use of popular tourism locations within the park is vital to assist the Komodo NP Management Authority formulate appropriate manta ray conservation and management policies.

Methods

This study uses a long-term library (MantaMatcher.org) of photo-identification data collected by researchers and citizen scientists to investigate manta ray demographics and habitat use within the park at four sites frequented by tour operators: Cauldron, Karang Makassar, Mawan, and Manta Alley. Residency and movements of manta rays were investigated with maximum likelihood analyses and Markov movement models.

Results

A total of 1,085 individual manta rays were identified from photographs dating from 2013 to 2018. In general, individual manta rays displayed a higher affinity to specific sites than others. The highest re-sighting probabilities came from the remote southern site, Manta Alley. Karang Makassar and Mawan are only ~5 km apart; however, manta rays displayed distinct site affinities. Exchange of individuals between Manta Alley and the two central sites (~35.5 km apart) occurred, particularly seasonally. More manta rays were recorded traveling from the south to the central area than vice versa. Female manta rays were more mobile than males. Similar demographic groups used Karang Makassar, Mawan, and Manta Alley for foraging, cleaning, cruising, or courtship activities. Conversely, a higher proportion of immature manta rays used the northern site, Cauldron, where foraging was commonly observed. Fishing gear-related injuries were noted on 56 individuals (~5%), and predatory injuries were present on 32 individuals (~3%). Tourism within the park increased from 2014 to 2017, with 34% more dive boats per survey at Karang Makassar and Mawan.

Discussion

The Komodo NP contains several distinct critical habitats for manta rays that encompass all demographics and accommodate seasonal manta ray movements. While the present study has not examined population trends, it does provide foundational data for such work. Continued research into manta ray abundance, long-range movements, and identifying and protecting other critical aggregation areas within the region is integral to securing the species’ recovery. We provide management recommendations to limit undue pressure on manta rays and their critical habitats from tourism.

Introduction

Even highly mobile marine megafauna species often spend a disproportionate amount of time in and return to specific sites within their broad range (e.g., Baird et al., 2008; Bowen & Karl, 2007; Dewar et al., 2008; Graham et al., 2016; Rooker et al., 2008). Identifying and protecting these habitats should be prioritized to maximize population recovery for threatened species (Chapman et al., 2015; Heupel, Carlson & Simpfendorfer, 2007; Hueter et al., 2005; Martins et al., 2018; Norse, 2005; Speed et al., 2010), particularly ecologically sensitive areas like nursery grounds, adult reproductive areas, and feeding grounds. It is also important to maintain protected movement corridors between such sites (Hooker et al., 2011).

Manta rays (Mobula alfredi–Krefft, 1868; M. birostris–Walbaum, 1792) are long-lived, to at an estimated 45 years (Marshall et al., 2019, 2020), with low fecundity of approximately one offspring every 2 years (Deakos, 2012; Marshall & Bennett, 2010), resulting in low population growth rates, estimated as maximum intrinsic rate of population increase (rmax), of 0.166 or less per year (Dulvy et al., 2014; Rambahiniarison et al., 2018). Both species are threatened with extinction (Marshall et al., 2019, 2020) due to declining population trends (Dulvy et al., 2014; Ward-Paige, Davis & Worm, 2013), and consequently are listed in Appendix II of the Convention on the International Trade of Endangered Species (CITES) and Appendices I and II of the Convention on the Conservation of Migratory Species (CMS). The demand for Mobula spp. branchial gill plates, used in Chinese non-traditional medicinal markets primarily located in Guangzhou, Macau, Hong Kong, and Singapore (Heinrichs et al., 2011), incentivizes fishers to target manta rays; this trade has become the most significant contributor to manta ray population declines globally (Marshall et al., 2019, 2020; O’Malley et al., 2017; Stewart et al., 2018a).

In the 1990s and early 2000s, manta rays were heavily fished in Indonesia, which ranked within the top five mobulid fisheries nations (Heinrichs et al., 2011). Manta landings declined substantially in the 2010s, suggesting that overexploitation was occurring, and potentially damaging the growing manta ray tourism industry (Booth, 2016; Lewis et al., 2015; O’Malley, Lee-Brooks & Medd, 2013). The Komodo National Park (NP), formally established in 1980, primarily for the conservation and management of the endemic Komodo dragon (Varanus komodoensis), was extended in 1984 to include surrounding marine habitats (Pet & Yeager, 2000). Coincidentally, the marine realm included habitats now known to be critical to Mobula alfredi (hereafter referred to as manta rays, as only this species is considered in this work). The first study on M. alfredi within Komodo NP (Dewar et al., 2008) predated the taxonomic split from M. birostris (Marshall, Compagno & Bennett, 2009) and investigated their movements within the area. Passive acoustic telemetry employed between October 2000 and June 2003 identified regular use of specific areas and the connectivity of manta ray sites within the park. These were welcome findings at a time when manta ray fisheries were active through the Lesser Sundas (Heinrichs et al., 2011; Lewis et al., 2015; White et al., 2006), as the ban on shark and ray catches, prohibition of the most detrimental and indiscriminate fishing gear (e.g. gillnets) within most of the park (Pet & Yeager, 2000) would afford the resident manta ray population an increased level of protection from fishers. The Komodo NP has subsequently been touted as a successful example of a beneficial Marine Protected Area (MPA) for manta rays (Rigby, Simpfendorfer & Cornish, 2019).

Following a sequence of manta ray conservation milestones (Setyawan et al., 2022), including Indonesian regional shark and manta fishing bans in Raja Ampat, West Papua (Raja Ampat Local Government, 2012) and West Manggarai, Flores and Komodo NP (West Manggarai Local Government, 2013), in 2014, both manta ray species were formally protected throughout the entire exclusive economic zone of Indonesia (an area of over 6 million km2; Dharmadi, Fahmi & Satria, 2015; Ministry of Marine Affairs & Fisheries, 2014). The nation-wide legislative protection of manta rays was adopted after these species were listed on the aforementioned conservation conventions, but the Indonesian government was also incentivized to protect the valuable manta ray tourism industry in the country, estimated to be worth over US$10.6 million per year (Mustika, Ichsan & Booth, 2020; O’Malley, Lee-Brooks & Medd, 2013). However, the species is still subject to bycatch and illegal directed fisheries (Booth et al., 2020; Croll et al., 2016), particularly in movement corridors to the south of Bali, Lombok, and Sumbawa islands (Conservation International, 2016; Germanov & Marshall, 2014; Lewis et al., 2015). Designated conservation areas, such as the Komodo NP and the Nusa Penida Marine Protected Area (MPA) can, at least in theory, offer an additional layer of protection due to fishing gear restrictions in marine protection and marine core zones and increased levels of compliance monitoring including from the tourism sector (Erdmann, 2004a; Komodo National Park Office, 2020a; IUCN World Heritage Outlook, 2020).

In the current study, we expand on earlier research in the Komodo NP (Dewar et al., 2008; Germanov & Marshall, 2014) by creating and analyzing a long-term photo-identification library compiled mostly during 5 years of extensive surveys to investigate habitat use of 1,104 identified manta rays. Our study goals are to assess: (1) site affinity and movement, (2) population structure, (3) behavior, and (4) persistent threats to manta rays using the surveyed sites within the Komodo NP and to identify specific recommendations to help mitigate these threats.

Materials and Methods

Study area

The Komodo National Park (Komodo NP; Fig. 1) covers 1,817 km2 of land and sea (Pet & Yeager, 2000; Erdmann, 2004b). This area lies east of Bali, south of Kalimantan and west of Papua, contains high levels of endemic terrestrial species and is on the southern side of the Flores Sea “Marine Wallace line” (Barber et al., 2000). This is a complex oceanographic region (Komodo NP marine area = 1,214 km2) characterized by three large islands (Komodo, Rinca, and Padar) and additional smaller islands within the park boundary totaling 603 km2 of terrestrial habitat (Erdmann, 2004b). The three large islands create the large shallow (≤100 m) Lintah Strait, through the park’s center. The park boundaries are flanked by Sape Strait, another large, slightly deeper (~100–200 m) strait to the west of Komodo Island, a narrow passage to the east of Rinca Island (Molo Strait), and deep water (>800 m) basins to the north and south (Fig. 1). Water exchange through the park is driven by strong tidal flow currents (up to ~15 km/h in the Lintah Strait), with water from the Indian Ocean flowing north through the straits on the incoming tide and water from the Pacific Ocean flowing south on the outgoing tide (Erdmann, 2004c) via the Indonesian Throughflow (ITF). Upwelling to the south of the Lesser Sundas is strongest from June through October, coinciding with the south-east monsoon, active May–October, and serves to enrich the waters in southern Komodo NP during this time of year (Ningsih, Rakhmaputeri & Harto, 2013). Nutrient-rich waters flow throughout the straits, particularly with spring tides. This trend is reversed during the inverse north-west monsoon period (November–April), with nutrient-rich water flowing through Komodo NP from the Flores Sea located to the north.

Figure 1 Location of study sites within the north (Cauldron–CL), central (Karang Makassar–KM, Mawan–MW), and south (Manta Alley–MA) regions of the Komodo National Park (NP), Lesser Sundas, Indonesia.

Core study sites (colored circles) and ‘other’ sites (white circles) within Komodo NP with documented manta ray sightings. These other location sites include (clockwise from north to south): ‘Batu Montjo’, ‘Castle’ and ‘Crystal Rock’, ‘Lighthouse’, ‘Tatawa Besar’, ‘Tatawa Kecil’, ‘Batu Bolong’, ‘Batu Sabun’, ‘Siaba Besar’, ‘Police Point’, ‘Padar Kecil’ and ‘German Flag’. The boundary of Komodo NP is highlighted by an orange line. The figure was created using QGis v 2.18, 2016 and bathymetry information was obtained from: GEBCO_2014 Grid, version 20150318; www.gebco.net.

We focused our analyses on four sites within the Komodo NP (from north to south: Cauldron–CL, Karang Makassar–KM, Mawan–MW, Manta Alley–MA) based on the consistent manta ray sightings at these locations and, hence, regular visitation by tourism operators, which facilitated both research and citizen science contributions. Three of these sites, KM, MW, and MA, were included in a previous acoustic telemetry study of manta ray movements within the park (Dewar et al., 2008). Cauldron, the previously undescribed site, located in the north area of the park, is a shallow channel (<25 m) between two islands with complex bathymetric structure and strongly affected by tidal currents. Located in the ‘central’ area of the park, adjacent to a sandy island with a fringing reef, KM refers to a gently sloping shallow (<18 m deep) rubble field that runs ~1.5 km north to south, with patchy reef and coral heads where manta rays clean (i.e., cleaning stations). The island MW, ~5 km to the east of KM, has a shallow ‘cleaning station’ at 5 m on a sandy slope from three down to 20 m on its south-eastern tip. Shallow rubble reefs also serve as cleaning stations to the north and south of MW’s sandy slope, covering an overall distance of ~0.7 km. Strong tidal currents affect both these central sites (i.e., KM and MW). In south-west Komodo NP, MA is approximately 35.5 km in a straight-line distance from KM. This site encompasses several rocky islands in a large bay lined with steep cliff walls and rocky shorelines. Several channel formations are exposed to surge and strong tidal current flows on the islands’ northernmost side, where manta rays clean and cruise. To the east and west of the islands are sloping reefs (to ~35 m), with several manta ray cleaning stations.

Data collection and processing

Data collection, including citizen science contributed data, validation, and processing, followed the procedures described in Germanov et al. (2019a). Data on manta ray sightings (date, time, location, and identifying ventral photographs of manta rays) logged by observers and the public were accessed from the online database MantaMatcher.org (Fig. S1). Approximately 20 local dive operator staff were trained by the authors as observers and contributed manta ray data from 2012–2018. Briefly, the training included providing trainees with details of how to take manta ray identification photographs, identify sex, maturity and behavior, estimate size, and to record injuries and entanglements with fishing gear, along with other data collection relevant to effective manta ray management in the region, such as the maximum number of dive boats present on-site throughout surveys (Germanov et al., 2019a). Photo contributions from the public were encouraged through educational presentations, informative dive briefings, and awareness materials about ‘Manta Matcher’ (i.e., posters and infographics) displayed at local dive centers and within ‘liveaboard’ dive boats.

All identifying photographs were manually matched to an ID catalog, with the assistance of an automated pattern matching algorithm (Germanov & Marshall, 2014; Town, Marshall & Sethasathien, 2013) or external software (‘MantaUtil,’ Winstanley, 2016). The lead author independently validated the sighting records included in the study. Manta ray sex was assigned based on the absence (female) or presence (male) of claspers, and maturity status was assigned based on clasper size in males, with those extending past the pelvic fins considered as mature, or the presence of a pregnancy bulge or pectoral fin mating scars in females (Marshall & Bennett, 2010). This methodology avoids a default of classifying females as immature in the absence of maturity indicators and accurate size estimates; instead, their maturity status was classed as “unknown” (Marshall & Bennett, 2010; Marshall, Dudgeon & Bennett, 2011). Behavior was classified into four mutually exclusive categories: foraging, cleaning, cruising, and courtship. Two additional non-exclusive behavior categories, foraging/cleaning, and courtship/cleaning were used when more than one behavior was observed for an individual manta ray within a single dive. Further details on behavioral categorization are provided in Germanov et al. (2019a).

The use of Manta Matcher also facilitated the identification of manta ray movements between geographical regions, namely re-sightings between Komodo NP and the Nusa Penida MPA. Sightings of all individuals are publicly available at MantaMatcher.org. Identification numbers for the most re-sighted individuals, those making long-range movements, and others with noteworthy observations are provided in the results section and the supplementary data. These identification numbers can be input into the online database’s search function to reveal full sighting histories for the individual manta rays.

Statistical analyses

The core sightings records used for statistical analyses (outlined below) were collected between January 2013 and April 2018. Annual logged sightings records exceeded 400 across those years, with near year-round coverage for the three of the four sites considered here. Data from 2018 were excluded from seasonal analyses, as year-round survey effort was not available, and from CL (all years) as there were relatively few records. The number of logged trained observer dives were used as a proxy for survey effort, assigned as per dive, as dive time was not recorded before 2016 (Fig. S2). However, dive times set by dive operators are a maximum of 60 min and the means of available dive times (post-2016) across sites were relatively consistent (58.3 ± 7.1 min). Since most dive operators complete two or more dives per day at the remote site MA (i.e., effectively doubling daily effort compared to other sites, E. Germanov, 2016, personal observation), we have presented total mean monthly sightings as a more appropriate comparison between MA and KM/MW as the number of hours and daily dives logged was not regularly available from public data.

Pearson’s product-moment correlations, using the cor.test function of the R statistical software (R Core Team, 2018), were used to investigate the relationship between the annual number of survey days and the number of sightings. The chisq.test function (R Core Team, 2018) was used to test whether the numbers of individuals and sightings differed between males and females at KM, MW, MA, and CL. Sex ratio data for each site were compared using chi-squared (χ2) goodness of fit tests (one-dimensional contingency table), while sex ratios and behavior frequencies between the sites were compared using a multiple-dimensional contingency table (VassarStats, 1998). To facilitate χ2 testing for behavior, where counts were less than five per site, the data were condensed into four categories (i.e., foraging/cleaning was reclassified to foraging, and courtship/cleaning was reclassified to courtship). Data from CL were excluded from χ2 testing for behavior, and data from both CL and MW were excluded from χ2 testing for seasonality in foraging, as the counts were less than five for several categories. A Fisher’s Exact Probability Test was used for 2 × 2 contingency tables if counts were less than five. The Yates’ continuity correction was applied to tests where there was one degree of freedom.

Residency and movement analyses

Following the habitat use definitions of Chapman et al. (2015), we use ‘residency’ to refer to the generally uninterrupted occupation of a limited area by an individual for a defined length of time, and ‘site fidelity’ as the return of an individual to a site after a periodic absence of greater or equal duration to the residency period. Presence-only sightings data precludes discerning whether these same site visitations are true site fidelity or, at least for some individuals, movements within a large home range. In this case, the term ‘site affinity’ is more appropriate for describing the same site re-sightings (Couturier et al., 2011). This term is also more appropriate in instances where there is high variability in site use between individuals and at least some individuals use several sites to a similar extent (see Germanov et al., 2019a).

We compared daily manta ray re-sighting data against residency models to investigate residency patterns using a modified maximum likelihood approach, following Germanov et al. (2019a). We excluded CL data from analyses, as sighting records at this site were sparse compared to those at KM, MW, and MA (Table 1). Lagged Identification Rate (LIR), defined as the probability of re-identifying an individual after a given time lag (Whitehead, 2001), was calculated using the ‘Movement Analyses’ module of the program SOCPROG 2.8 (Whitehead, 2009). Empirical results were compared to model closed and open populations scenarios to estimate movement parameters (Table 2 and Table S1). We evaluated emigration, immigration, re-immigration, and mortality in the various open population models. The lowest quasi-Akaike information criterion (QAIC) value, accounting for the over-dispersion of the data, determined the model that best fit the residency characteristics for each site (Whitehead, 2007). We estimated the probability of re-sightings at other sites using ‘within/between’ LIR analysis to test for population-level mixing between sites. Model fits were bootstrapped 1,000 times to generate standard errors (SE).

Table 1 Individual manta rays, sightings, and survey days between July 2004 and April 2018.

Data are reported for Komodo National Park (KNP) overall, Cauldron (CL), Karang Makassar (KM), Mawan (MW), Manta Alley (MA), other sites within KNP and for manta rays sighted in KNP as well as in the Nusa Penida Marine Protected Area (NP MPA). Daily duplicates are removed (n = 755).

	All KNP	CL	KM	MW	MA	Other KNP	NP MPA	
Year				Individuals				
2004–12	139	1	122	–	16	2	5	
2013	308	3	186	58	103	3	4	
2014	473	8	270	92	186	5	6	
2015	371	5	192	127	102	2	7	
2016	457	8	277	103	159	9	8	
2017	688	34	354	348	116	24	4	
2018*	167	1	103	72	–	1	1	
TOTAL	1,104	48	779	535	383	45	11	
				Sightings				
2004–12	177	1	156	-	18	2	13	
2013	463	3	246	67	144	3	19	
2014	838	19	404	116	294	5	16	
2015	535	11	229	147	146	2	13	
2016	764	10	375	117	253	9	19	
2017	1,198	59	476	464	174	25	6	
2018*	188	1	107	79	-	1	3	
TOTAL	4,163	104	1,993	990	1,029	47	89	
				Survey days				
2004–12	73	1	66	-	8	2	13	
2013	116	2	85	23	14	3	19	
2014	150	12	94	39	39	3	15	
2015	97	9	56	35	15	2	12	
2016	114	7	77	33	21	9	18	
2017	198	30	103	103	13	20	6	
2018*	43	1	26	25	-	1	3	
TOTAL	791	62	507	258	110	40	86	
								
Notes:

* To April only.

- = no data available for this time.

Table 2 Residency model parameters and fits (ΔQAIC) for individual manta ray sightings at Karang Makassar (KM), Mawan (MW), and Manta Alley (MA), Komodo National Park.

	Study sites	KM	MW	MA	
Model	Model description	ΔQAIC	ΔQAIC	ΔQAIC	
A	Closed (1/a1 = N)	58,178.48	13,966.47	81.84	
B	Closed (a1 = N)	101.70	10.14	81.84	
C	Emigration/mortality (a1 = emigration rate; 1/a2 = N)	15.77	4.83	11.90	
D	Emigration/mortality (a1 = N; a2 = mean residence)	15.77	4.83	11.90	
E	Closed: emigration + re-immigration (a1 = emigration rate; a2/(a2 + a3) = proportion of population in study area at any time)	73.33	3.12	61.46	
F	Emigration + re-immigration (a1 = N; a2 = res time in; a3 = res time out)	42.88	3.12	2.90	
G	Emigration + re-immigration + mortality	1,547.52	8.95	781.20	
H	Emigration + re-immigration + mortality (a1 = N; a2 = res time in; a3 = res time out; a4 = mort)	0	0	0	
Note:

N = population.

The annual transition probabilities between sites, i.e., the likelihood of an individual manta ray moving from one area to another within a year, were calculated using a parameterized Markov movement model (Tables 3 and 4; Whitehead, 2009). This model includes a hypothetical ‘outside’ area (i.e., leaving the study site/s). Movements between the three core sites (KM, MW, MA) were investigated by grouping KM and MW as a collective ‘central’ site and investigating movements to and from MA in the south. Optimized values of transition probabilities were bootstrapped 1,000 times to generate SEs, and the maximum number of evaluations was set to 10,000. Mortality, including permanent emigration from all core sites, was considered in the model.

Table 3 The estimated probability (±1 SE) of re-sighting an individual manta ray in the same or another site.

Movement probabilities are presented for manta rays (n = 1,061) at (A) the three Komodo NP core sites Karang Makassar (KM), Mawan (MW), Manta Alley (MA) and additional ‘outside’ sites. (B) Sites grouped into central (KM + MW) and south (MA) regions.

	To:	KM	MW	MA	Outside	
A) Core
From:	KM	0.39	0.37 ± 0.16	0.13 ± 0.03	0.11 ± 0.07	
MW	0.29 ± 0.17	0.59	0.01 ± 0.02	0.02 ± 0.05	
MA	0.20 ± 0.97	0.23 ± 0.06	0.58	0.00 ± 0.03	
	Outside	0.10 ± 0.06	0.12 ± 0.05	0.00 ± 0.03	0.79	
		To:	Central	South	Outside	
B) Central, south
From:	Central	0.87	0.11 ± 0.04	0.03 ± 0.20	
		South	0.47 ± 0.11	0.53	0.03 ± 0.01	
		Outside	0.11 ± 0.20	0.00 ± 0.02	0.89	

Table 4 The estimated probability (±1 SE) of re-sighting an individual manta ray in the same or another site according to sex.

Movement probabilities are presented for (A) males and (B) females at core sites Karang Makassar (KM), Mawan (MW) and Manta Alley (MA), Komodo National Park.

From area:	To area:	KM	MW	MA	Outside	
A) Males	KM	0.53	0.27 ± 0.15	0.11 ± 0.02	0.08 ± 0.05	
(n = 507)	MW	0.32 ± 0.12	0.51	0.08 ± 0.02	0.10 ± 0.09	
	MA	0.21 ± 0.07	0.15 ± 0.06	0.64	0.00 ± 0.00	
	Outside	0.02 ± 0.02	0.02 ± 0.02	0.00 ± 0.00	0.96	
B) Females	KM	0.43	0.28 ± 0.17	0.17 ± 0.05	0.11 ± 0.01	
(n = 498)	MW	0.32 ± 0.18	0.41	0.15 ± 0.04	0.02 ± 0.05	
	MA	0.23 ± 0.11	0.19 ± 0.10	0.58	0.00 ± 0.00	
	Outside	0.15 ± 0.09	0.09 ± 0.04	0.00 ± 0.00	0.79	

Ethics statement

This study was conducted under permits issued by the Indonesian Ministry of Research and Technology (Permit #458/SIP/FRP/E5/Dit. KI/XII/2015; Permit Extension#11/TKPIPA/E5/Dit. KI/XI/2016 and #86/EXT/SIP/FRP/E5/Dit.KI/XI/2017) and the Komodo National Park (#SI.1432/BTNK-1/2016 and 2017). This study was carried out in accordance with the approval of the Animal Ethics Committee, Murdoch University (R2781/15). Photographic and sighting data were collected opportunistically by the public and contributed to a public online repository (MantaMatcher.org) developed explicitly to facilitate citizen science contributions to manta ray research.

Results

Sightings and survey effort

We identified a total of 1,104 individual manta rays from 4,163 sightings (after excluding 765 daily re-sightings of the same individuals) across 791 unique dates (from 7 July 2004 to 31 March 2018) within the Komodo NP (Table 1). Sighting records for four sites within the park, Cauldron (CL), Karang Makassar (KM), Mawan (MW), and Manta Alley (MA), and within core survey years (2013–2018) represented ~94.7% (n = 3,941) of all sightings (Table 1). Sightings records prior to 2013 (n = 177) included 139 individuals. Annual manta ray sightings varied across study years (Table 1) and were positively correlated with the number of survey days (r = 0.94, p = 0.002). More than 400 sightings were logged on Manta Matcher in each of the core years of this study (Table 1), with citizen science contributions representing 95% of all data logged. Data submissions to Manta Matcher gradually increased over the study period with increased awareness of the citizen science program within the dive community, surpassing data collected solely by trained observers in the second year of intensive surveys (i.e. 2014; Fig. S1).

A total of 1,085 individual manta rays were identified from data collected during the core study years (January 2013 to April 2018) from 14 sites within the Komodo NP. Data collected from the three core study sites KM, MW, MA, yielded 1,061 individual manta rays. Of these individuals, 749 were sighted in KM, 535 in MW, and 376 in MA. Data collected from CL yielded 48 individuals, of which 22 were not sighted elsewhere, and two were sighted at all of the other three sites. A further 45 individual manta rays also had sighting records in sites other than the above four sites, of which two had no other records elsewhere. Considering KM, MW, and MA sites combined (i.e., the three sites with 97.8% of all individual records from 2013–2018), the discovery curves of newly discovered individuals and the days elapsed showed a steep rise until approximately 685 individuals after 860 d (Fig. 2). The curve continued to increase at a slightly lower rate until the end of the study period (1,061 individuals after 1,913 d) (Fig. 2). Neither the combined discovery curve nor any of the site-specific discovery curves approached an asymptote. Gaps in the discovery curves indicate periods in which no new individuals were identified due to survey gaps.

Figure 2 Discovery curves for newly-identified manta rays in Komodo National Park over time (in days) from January 2013 until April 2018.

Discovery curves are presented for the sites combined (Comb.) and separately for Karang Makassar (KM), Mawan (MW), and Manta Alley (MA). NS, number of survey days; NI, number of individuals.

Sightings across seasons and years

Sightings rates averaged (mean ± 1SE) 760 ± 130 sightings/year from 2013–2017, and January–March of 2018 had 188 sighting records. The highest number of sighting records was recorded in 2017 (n = 1,198), with the second-highest number in 2014 (n = 838; Table 1). However, survey effort by trained observers was lower in 2015 and 2016 (Fig. S1). Monthly mean sightings varied among years (Fig. 3A). There was an overall increase in sightings from 2013 to 2017 across the sites, with the largest increase observed at MW. Sightings per dive recorded solely by trained observers varied modestly for sites among years, except for KM, where there was a ~two-fold increase in sightings per dive during 2015 and 2016 (Fig. S2A). Sightings per dive by trained observers also varied across months and sites (Fig. S2B), following similar trends to the sightings overall, with fewer sightings for the central sites (MW and KM) in August and September (Fig. 3B). While trained observers recorded higher sighting rates for MA mid-year, data were not available for December–February and in June, prohibiting us from commenting on seasonality trends based on these data alone.

Figure 3 Total mean monthly sightings (±1 SE) of identified manta rays for the Komodo National Park at Karang Makassar (KM), Mawan (MW) and Manta Alley (MA) from 2013–2017.

Data are presented as annual (A) and monthly (B) mean sightings per month. KM = 1,730 sightings, 415 days; MW = 911 sightings, 233 days; MA = 1,011 sightings, 102 days.

Population structure

Of 1,061 individuals recorded at the core sites (KM, MW, MA), the overall sex ratio of males (507) to females (498) of 1.02:1 did not differ significantly from 1:1 (χ21 = 0.1, p = 0.777), with the sex of 56 individuals remaining unknown during the core study years. Likewise, the sex ratio did not differ significantly from 1:1 at any of the three core sites: KM (0.92:1, χ21 = 1.4, p = 0.234), MW (0.97:1, χ21 = 0.2, p = 0.692) or MA (0.9:1, χ21 = 1.0, p = 0.319) (Fig. 4). While there were more males than females (23 vs. 16, with nine unknown individuals) identified at CL, this difference was not significant (χ21 = 1.3, p = 0.262). There was no significant association between site and sex for individuals (χ22 = 0.3, p = 0.851) at the three core sites.

Figure 4 Population structure of manta rays in the Komodo National Park.

The proportion of each demographic group is given as a percent of the individuals sighted in each site (from north to south): Cauldron (CL), Karang Makassar (KM), Mawan (MW), and Manta Alley (MA). The records are from January 2013 to April 2018.

Approximately 90% (454) of the males at the core study sites were sexually mature, with ~11% (50) of males remaining immature during the five core study years. An additional 11 immature males were identified at CL and not sighted at other sites. Altogether, 96 (19%) males were immature at some point throughout the study, with 39 reaching maturity over this period. Females are more difficult to assign a maturity status without accurate size estimates, which the study lacked, and for 290 females representing ~27% of individuals, the maturity status was unknown. Considering KM, MW and MA combined (n = 498 females), at least 43% (212) of the females were sexually mature, and 18% (92) were pregnant during the study period. Overall, the maturity status of males and females was comparable among all sites for individuals (Fig. 4; χ23 = 1.2, p = 0.742).

However, looking at maturity status alone (i.e., excluding sex), there was a relationship between the site and maturity status of individuals (χ23 = 78.7, p < 0.0001), with relatively higher composition of immature males (31%) and lower composition of mature females (4%) at CL compared to the other three sites (Fig. 4). Notably, of the manta rays identified at CL sighted more than once (4.29 ± 0.59; range: 2–11 sightings), seven out of 17 individuals (41%) were immature males, with two maturing during the study (INKNP0598A and INKNP0292A; Fig. S3). The maturity status of the remaining eight manta rays encountered more than once at CL was unknown (six females and two where the sex was unknown). In contrast, all manta rays (females = 6 and males = 5; Fig. S4) with recorded long-range movements between Komodo NP and Nusa Penida MPA were deemed mature.

Residency and movement

From 2013–2018, the majority (768; 72%) of manta rays were encountered more than once within KM, MW, and MA (Fig. S5). Of these individuals, 48 (5%) were sighted >10 times, with up to a maximum of 21 sightings per individual. The mean re-sightings per individual were 3.6 ± 0.1, and 598 manta rays (56%) were re-sighted across multiple years (Fig. 5). Considering all available data (2004–2018), the longest time between the initial and most current re-sighting for an individual (INKNP0154A) was 13.4 years. Further, using all available data (2004–2018) on long-range movements to Nusa Penida MPA, the days between recaptures in the different regions for the 11 individuals varied widely (range: 33–1,550 d), averaging 373.4 ± 77.4 (SE) d. In addition to sightings within the Komodo NP, 11 (1%) individual manta rays sighted within the Komodo NP were also sighted within the Nusa Penida MPA on 89 separate days (including pre-2013 data; Fig. S4).

Figure 5 Individual manta ray (N = 1,061) sighting span (years) from the core study sites combined.

Core sites are Karang Makassar, Mawan and Manta Alley. The records are from January 2013 to April 2018.

Interchange between sites

We recorded 104 (9.8%) individuals in all three core sites, and 275 (25.9%) in two sites, one being a central site (either KM or MW) and the other MA in the south. We recorded 961 (91%) individuals in the park’s central sites and an additional 100 (9%) individuals exclusively at MA. The Markov movement model showed that each individual’s re-sighting probability was highest within rather than between sites (0.39 at KM, 0.59 at MW, and 0.58 at MA; Table 3A). However, there was a relatively high probability of movement from KM to MW (0.37 ± 0.16) or vice versa (0.29 ± 0.17), indicating a sizeable interchange between the two adjacent central sites (~5 km apart). When these two central sites were grouped for analysis (Table 3B), the probability of re-sighting within the central area was very high (0.87), with less, yet considerable re-sightings (0.53) at MA. Further, the movement between central sites and MA was asymmetric, with far less movement from central sites to MA (0.11 ± 0.04) than vice versa (0.47 ± 0.11). More movement to the central sites from other ‘outside’ sites was modeled (0.11 ± 0.20) than from ‘outside’ sites to MA (0.000 ± 0.02). Notably, the model’s maximum number of iterations was exceeded for these analyses, suggesting that the estimated SEs are inaccurate and larger than expected (H. Whitehead, Dalhousie University, 2019, personal communication). While excluded from movement analyses due to the smaller sample size, 40% of individuals sighted at CL were also sighted at central sites, and 13% at MA.

We tested for full interchange between sites using LIR analysis (Fig. 6). The curves for re-sightings within the same or different sites did not converge during the study period, indicating that some level of site affinity exists. A ‘within/ between’ analysis between just KM and MW also did not support full interchange within the central region (Fig. S6). Thus we ran LIRs for each site independently.

Figure 6 The probability of an individual manta ray being identified in the same or a different site within Komodo National Park over time (days).

Lagged Identification Rates (LIR) (± SE) were calculated with records from January 2013 to April 2018.

Site-specific analyses

The best-fit re-sighting model for all three sites was Model H (Table 2), a model which includes emigration and re-immigration with mortality. Mortality (which includes permanent emigration) was considered negligible for all sites and analyses (≤0.045). The LIRs within the two central sites were similar and much lower (~half) than those at MA, indicating that individuals have a higher probability of re-sightings at MA than either KM or MW (Fig. 7). Model H scenarios provide estimates (mean ± SE) for the time individuals spend within (residence time in) and out (residence time out) of an area. Individuals stayed approximately twice as long at MA (1.6 ± 12.9 d) than at MW (0.8 ± 0.3 d), although there was high variability in residence time among individuals. The estimated time that individuals spent outside of these three sites ranged between 5.8–6.9 d. Overall, the results for MA indicated more variability between individuals’ residence time than at MW and KM (Table S2).

Figure 7 The probability of an individual manta ray being identified over time (days) within each of the three core sites.

Lagged Identification Rates (LIR) (± SE) for the three core study areas in Komodo NP: Karang Makassar (KM), Mawan (MW), and Manta Alley (MA), Komodo NP were calculated with records from January 2013 to April 2018. Best-fit LIR models (dotted lines) are shown for each site. Standard Errors (SE) are depicted as vertical lines for each data point.

Sex-linked analyses

There were significantly more sightings of females (2,016) than males (1,672) (χ21 = 32.1, p < 0.0001). This was the trend observed at KM, MW, and MA, with KM having the highest female sightings (57%) compared to males (43%) (χ21 = 35.1, p < 0.0001). In contrast, the comparatively fewer (103) sightings at CL were biased towards males (66% vs. 34% females, χ21 = 9.8, p = 0.002). Likewise to the population structural differences outlined above, there was no significant association between site and sex for sightings (χ22 = 4.9, p = 0.087) at the three core sites. However, when CL was included in the analysis, an association was detected (χ23 = 21.5, p < 0.0001). Exploration into potential sex-linked differences in site use showed no substantial differences in LIRs between the sexes for KM, MW, or MA (Fig. S7, Table S1). Movement analyses indicated that females tended to move more from central sites to MA, and from ‘outside’ to the central sites, than males (Table 4), suggesting that there might be some sex-linked differences in site use. Adequate data was not available to perform LIR or movement analyses with sightings at CL.

Behavior

Manta rays were observed foraging, cleaning, cruising, and engaging in courtship at KM, MW, and MA (n = 2,306, Fig. 8A). There was a significant association between behavior and site (χ26 = 202.1, p < 0.001), with MW having the highest proportion of cleaning behavior (74%), while CL had the lowest (6%). More cleaning was observed at KM (53%) than MA (39%). Overall, few observations of mixed behaviors (i.e., cleaning and courtship = 20; cleaning and foraging = 1) were made for individuals within a single survey dive. Courtship behavior was observed at all sites except CL. Courtship activity took place throughout the year, although the number of recorded events and individuals engaged in courtship varied (Fig. 8B). Cruising behavior was frequently observed at CL (74%) and, to a lesser extent, MA (45%). Both locations have strong currents flowing through narrow channels where manta rays are commonly observed swimming into the current, but not necessarily foraging or cleaning. Foraging observations differed between the study sites. There were fewer foraging records for individuals at MW (1%) than at other sites, especially CL, where foraging records were higher than elsewhere (20%). The highest number of individuals foraging at one time (up to 30 individual identifications in 1 day) were recorded at KM during the NW monsoon.

Figure 8 Manta ray behavioral habitat use and seasonality trends in the Komodo National Park.

(A) Manta ray behaviors at four sites: Cauldron (CL, n = 49), Karang Makassar (KM, n = 1,067), Mawan (MW, n = 741) and Manta Alley (MA, n = 498). The records are from January 2013 to April 2018. (B) The average identified manta rays engaging in courtship behavior on a daily basis broken down by month. The data was recorded between 2013 and 2017 across sites KM, MW, and MA and presented as the daily means (±1 SE). The numbers above the month bars indicate the number of courtship events recorded. The shaded and white backgrounds indicate the north-west monsoon (NW; November–April) and south-east monsoon (SE; May–October), respectively.

Boating activity

The two central sites, KM and MW, are the closest to the population center at Labuan Bajo on Flores Island (Fig. 1), from where most diving/snorkeling tours operate. Based on observer logs, these sites saw a combined 34% increase in the number of tour boats per survey recorded from 2014 to 2017. This increase was proportionally greater at MW than at KM, with the survey average of boats at MW increasing from 1.8 in 2014 to 3.4 in 2017 (81% increase), while at KM, boats increased from 5.8 in 2014 to 7.7 in 2017 (33% increase; Fig. 9A). The number of boats on-site during July or August surveys were 102% (12.3 boats) and 50% higher (4.1 boats) than the September–June average for KM and MW, respectively (Fig. 9B). Records for CL and MA were sparse and were not available year-round. However, for comparative purposes, and to establish a baseline, from 2014–2017 inclusive, the survey mean and median numbers of boats at CL were 2.6 ± 0.3 and 2 (range: 1–14), respectively (n = 57). While the number of boats is much lower than those of the central sites, the daily numbers in August 2017 were well above average (i.e., 10 and 14 boats), suggesting that a significant increase in tourism is also occurring at this site. Based on limited records (n = 27), it appears that the per survey mean (1.9 ± 0.2) and median (2; range: 1–4) boat numbers at MA have remained relatively stable over time.

Figure 9 Annual and monthly variation in the survey average (±1 SE) number of boats present at Komodo National Park manta ray sites Karang Makassar (KM) and Mawan (MW).

Records from 2014 and 2017 are presented as annual (A) and monthly (B) mean counts per survey.

Injury rates

Predatory injuries, i.e., bite marks, were present on 32 (~3%) individuals. Fifty-six individuals (~5%) had cephalic fin, pectoral fin, or fishing line injuries. Nine individuals had more than one injury. The breakdown of injuries was: 13 individuals with hook and line entanglements and 30 and 26 truncations or disfigurements to the cephalic fins and pectoral fins, respectively. We noted six injured pregnant individuals during the study period.

Discussion

A substantial number of manta rays display site affinity to the Komodo National Park (NP), noteworthy relative to other well-studied manta ray habitats globally (compiled in Germanov et al., 2019a; Setyawan et al., 2020; Stevens, 2016). New manta ray identifications were common throughout the study, with a steadily increasing discovery curve of ~1,100 individuals, that will likely increase with continued survey effort. The present study is the most comprehensive study on manta rays residing in Komodo NP and characterizes site affinity, demographics, and behaviors in aggregation sites popular with tourism. While individual manta rays are highly mobile (Germanov & Marshall, 2014) and 1% of individuals (n = 11) have been re-sighted ~450 km away at the Nusa Penida MPA, they show distinct site affinity within the Komodo NP. Manta rays were commonly sighted at four sites in the park: Karang Makassar (KM), Mawan (MW), Manta Alley (MA), and Cauldron (CL). Site use differed between demographic groups, with mature individuals frequenting the reefs for cleaning and the initiation of courtship activity and immature males frequenting sites where foraging activity commonly occurs. The heterogeneity of site use indicates management requirements need to be tailored to prevent disruption to manta ray behaviors.

Long-term trends in site use and movement conserved in Komodo NP

More than 1,100 manta rays were re-sighted from 14 sites within the Komodo NP and displayed a high degree of site affinity to distinct locations. While the results from our study indicate that connectivity exists throughout the park, the study population was not fully mixed. Further, affinity within the park was not uniform, with manta rays displaying higher affinity to MA in the south than central areas in the park (MW, KW). These observations parallel the conclusions of an earlier acoustic telemetry study using sighting-independent monitoring and indicate that residency trends were conserved for over a decade (Dewar et al., 2008). Long-term high site affinity has been reported for manta rays elsewhere in Indonesia (Germanov et al., 2019a; Setyawan et al., 2020) and globally (Andrzejaczek et al., 2020; Couturier et al., 2018; van Duinkerken, 2010; Peel et al., 2019; Venables et al., 2020). Together, these similar findings across the years and geographies indicate that even moderately sized conservation areas (~2,000 km2) such as the Komodo NP, when they encompass critical habitats and complete demographics, can likely afford long-term protection to resident manta rays. Conversely, there will be diminished conservation benefits to individuals with high site affinity to sites outside conservation areas (Stewart et al., 2016). Therefore, the nationwide manta ray fisheries ban in Indonesia is a step in the right direction. However, to continue to improve manta ray conservation, an understanding of where all key aggregations occur is necessary, and additional conservation measures, such as banning destructive and indiscriminate fishing practices in these areas should be considered.

Estimates of residence time in and out of sites were most variable for MA, potentially a reflection of differential site use by a specific demographic; e.g., males being more mobile than females (van Duinkerken, 2010), mature individuals being more mobile than juveniles (Germanov et al., 2019a; Peel et al., 2019), or social groups forming (Perryman et al., 2019). Individual variation in residency and regional movements of manta rays was also observed in Raja Ampat, Indonesia (Setyawan et al., 2018) and the British Indian Ocean territory (Andrzejaczek et al., 2020). Notably, Dewar et al. (2008) also reported site preferences among individuals, with site visitations often corresponding to tagging locations. Further, the previous study recorded frequent observations of manta rays feeding at German Flag, a site directly adjacent to MA (~1.8 km away). The present study lacked information from German Flag but, based on the results from Dewar et al. (2008), it is possible that sustained foraging opportunities at this site contributed to the higher residency at MA, particularly as in the previous study, 20 out of 24 individuals (83.3%) recorded at German Flag were also recorded at MA.

There was higher interchange (i.e., immigration and emigration) between the two nearby ‘central’ sites (~4.5 km apart) than elsewhere, indicating substantial connectivity between the individuals using these locations. More individuals were identified from the two central sites, but these individuals had a lower probability of being re-sighted (lower LIR) than those observed at MA. Further, the resident times in central sites were also shorter than those observed at MA, albeit with less variability. These observations could be explained by repeated movements of individuals between the two central sites. Indeed, grouping KM and MW for movement analysis indicated that most movements were within this central area rather than to MA in the south-west. However, a certain degree of structure was still maintained. Conversely, nearly half of the individuals moved northward from MA to the central area of the park. A further notable percentage (~11%) of movements to the central sites were from ‘outside’ areas. These outside areas include CL, other sites within (e.g. Padar Kecil) and adjacent to the park (e.g. the island of Gili Banta), and farther manta ray aggregation areas, such as the Nusa Penida MPA (Dewar et al., 2008; Germanov & Marshall, 2014; Conservation International, 2016).

A notable difference between the present study and the previous acoustic telemetry study (Dewar et al., 2008) is that MW in central Komodo NP contained an important cleaning station that was visited by half of the region’s known individuals. In contrast, few visitations to this site were previously reported despite >1-year receiver deployment. The previous study’s observations could be an artifact of a single manta being tagged at this location, given the site affinity observed in the present study. Further, the northern site CL, which this study highlights as an important feeding area for immature manta rays, was not previously reported in Dewar et al. (2008). In the present study, the few sighting records from German Flag, and none at Padar Kecil, which the previous study identified as important sites, likely result from insufficient sampling effort at these sites.

Our sighting-based approach was limited temporally and spatially to when and where divers are present in the water and manta rays were photographed (i.e., by observer/photographer effort). For example, annual weather patterns influenced the results. Strong winds during December–February limit access to the north (CL) and central (KM, MW) sites, and there is limited access to the south (MA) during June–August. Environmental conditions consequently reduced survey effort. The few manta ray sightings recorded from December–March for MA might also be an artifact of low sampling effort, as most boats do not travel to MA at that time. However, Dewar et al. (2008) also reported fewer manta visitations to MA during December–March via a sightings-independent acoustic telemetry approach. Further, few tour operators regularly visit Padar Kecil and German Flag because of suboptimal environmental conditions at these sites (e.g., swell and low visibility). Nevertheless, the similarity in results between the present study and Dewar et al. (2008) validates the use of sightings data and a citizen science approach to make broad site use and movement observations that complement traditional telemetry approaches.

Habitats for a range of manta ray demographics and behaviors

All manta ray demographic groups were represented within Komodo NP. The three core sites, KM, MW, and MA, all had similar demographic profiles dominated by mature manta rays. Komodo NP’s manta rays had an equal ratio of males to females overall, in contrast to many other well-studied populations (Germanov et al., 2019a; Setyawan et al., 2020). However, the observed bias towards mature males over females might be at least partially attributed to the difficulty in assigning maturity to females in the absence of accurate size estimates and visual maturity indicators (i.e., mating wing scars or pregnancy bulge). At sites where mature manta rays were prevalent (i.e., KM, MW, and MA), cleaning was generally the most observed behavior, and courtship activity was only observed at these sites. These observations are like those from Manta Point in the Nusa Penida MPA, where the site is used predominantly for cleaning, social activity, and courtship (Germanov et al., 2019a), as well as observations at reefs within Raja Ampat that were deemed to be “courtship super sites” (Setyawan et al., 2020). Our findings support the concept that cleaning stations act as lekking sites i.e., aggregation areas where courtship and mate competition take place (Deakos, Baker & Bejder, 2011; Stevens, 2016). Dewar et al. (2008) revealed that manta rays typically used habitats nearby receivers within Komodo NP during the day, indicating that our sightings-based sampling during the day is an accurate record of site use. However, manta rays using deep cleaning stations, such as at MA, limit survey time due to no-decompression recreational diving limits. Further, manta rays feeding, cruising, and cleaning in strong currents and large swells are difficult to photograph, potentially influencing the study results.

Similar age class designations between the present study and a previous one in Nusa Penida MPA allow demographic comparison between the two areas. We noted fewer immature individuals were recorded in Komodo NP (n = 96, 8.7%) than Nusa Penida MPA (n = 123, 19.7%; Germanov et al., 2019a). This discrepancy in the identified immature individuals in Komodo NP (as compared to Nusa Penida MPA) likely indicates insufficient survey effort at CL and at other sites not regularly visited by tour operators (e.g., German Flag and Batu Montjo, Padar Kecil, and Gili Banta, located on the outside the park) that may be important to immature individuals (M. Erdmann, Conservation International, 2021, personal communication) Of the sites studied, we observed proportionally more immature male rays, fewer visibly mature female rays, and foraging behaviors at CL. Several manta ray studies have recently proposed several manta ray aggregation sites as potential nurseries (e.g. Germanov et al., 2019a; McCauley et al., 2014; Pate & Marshall, 2020; Setyawan et al., 2020; Stevens, 2016; Stewart et al., 2018b). Elasmobranch nurseries are designated according to the currently recognized criteria as habitats where young-of-the-year (YOY) are (1) disproportionately sighted compared to adults and are used for (2) extended periods and (3) across years (Heupel, Carlson & Simpfendorfer, 2007; Martins et al., 2018). These habitats likely provide substantial benefits to maturing young, such as nutrition, temperature regulation, and predator avoidance (Heupel et al., 2018). We did not have the data to discern YOY from generally immature individuals, an important distinction when considering nursery habitats, nor sufficient continued survey effort at CL, and thus are currently unable to test the nursery criteria. However, we note that the limited data available indicates that compared to other studied sites in Komodo NP, the majority of manta rays using the site were not mature (criteria 1), immature manta rays were repeatedly sighted for several months (criteria 2), and immature individuals consistently used the site across years (criteria 3). To definitively confirm CL as a nursery requires accurate measurements of the manta rays using the site and robust site use assessments, such as tracking studies to test criteria 2 (Heupel et al., 2018). In addition to CL, it would be beneficial to study several other sites within Komodo NP where immature individuals were reported for potential nursery areas. These sites include shallow protected bays similar to proposed nurseries elsewhere (Germanov et al., 2019a; McCauley et al., 2014; Pate & Marshall, 2020; Setyawan et al., 2020; Stevens, 2016; Stewart et al., 2018b), including Gili Banta (outside the Park), Batu Montjo and Lighthouse in the north and German Flag in the south.

Long-term observations of immature males within Komodo NP revealed that 50 (52%) did not reach maturity within the five core study years, whereas 39 did (40%). The age of maturity for male manta rays reported ranges from 3 to 13 years, depending on the study location (Stewart et al., 2018a). Our observations suggest that the age of maturity for male manta rays in Komodo NP might be closer to those reported for Nusa Penida, Indonesia (3–6 yrs; Germanov et al., 2019a), Raja Ampat, Indonesia (3–6 yrs; Setyawan et al., 2020), Mozambique (3–6 yrs; Marshall, Dudgeon & Bennett, 2011) and Japan (4–9 yrs; Kashiwagi, 2014) than in the Maldives (9–13 yrs; Stevens, 2016). However, the lack of size data precludes more detailed analyses on the age of maturity. Population doubling times for M. alfredi have not been estimated; however, estimates for M. birostris indicate long population doubling times of ~15–87 years (Rambahiniarison et al., 2018), and the estimated generation length for both species is 29 years (Marshall et al., 2019, 2020). Thus, long-term, ongoing studies are crucial to ensure current management and protection strategies are effective and adaptive (Pet & Yeager, 2000), and provide data to answer key questions on manta ray population dynamics.

Demographic influences on site use and movements

Population demographics, particularly age- and sex-linked differences in movements, appear to influence fine-scale habitat use in manta rays (Deakos, Baker & Bejder, 2011;van Duinkerken, 2010; Germanov et al., 2019a; Perryman et al., 2019; Stevens, 2016). However, in Komodo NP, unlike in the nearest studied aggregation in Nusa Penida MPA (Germanov et al., 2019a), the Lagged Identification Rates—LIRs did not differ between the sexes. Similarly, Couturier et al. (2018) did not consider sex a significant predictor of manta ray visitation rates to Lady Elliot Island in Australia. However, movement analysis suggests that females within the Komodo NP are slightly more mobile than males, and females may have ranged ‘outside’ the study sites more than males. In contrast, studies elsewhere report that males are more mobile than females (Deakos, Baker & Bejder, 2011; van Duinkerken, 2010; Germanov et al., 2019a; Perryman et al., 2019; Stevens, 2016). However, sex-biased dispersal does not appear to be the sole factor in the long-range movements from Komodo NP to the Nusa Penida MPA, as individuals of both sexes were documented moving between these locations (Germanov & Marshall, 2014).

Once mature, it appears that individuals typically shift to a more mobile lifestyle, characterized by increased movements between nearby sites in Komodo NP, Nusa Penida MPA (Germanov et al., 2019a), and elsewhere (Peel et al., 2019). Within the Nusa Penida MPA (Germanov et al., 2019a), we suggested that age-linked shifts in site use were linked to prey availability and prey density (Armstrong et al., 2016). Thus, a lower prey density might be able to sustain immature or smaller individuals, whereas larger individuals might need to forage primarily offshore to meet their greater energy demands (Lawson et al., 2019; Nøttestad et al., 1999). Another potential explanation for age-linked shifts in site use is the formation of social structures between individuals, which appear to be demographically influenced in manta rays (Perryman et al., 2019) and other elasmobranchs (Heupel & Simpfendorfer, 2005; Guttridge et al., 2011).

Environmental influence on seasonal movements

The Komodo NP is uniquely located in an area of several oceanographic features (i.e., mixing of waters from the Pacific Ocean via the Indonesian Throughflow (ITF) and the Indian Ocean, with tidal mixing, and productive upwelling) which are likely to provide highly productive waters year-round for manta rays (Ningsih, Rakhmaputeri & Harto, 2013). The local productivity and zooplankton availability in the region shifts in response to these broader regional oceanographic processes. Densely concentrated prey makes for prime feeding grounds to support large manta ray aggregations (Armstrong et al., 2016), and similarly to the Maldives (Anderson, Adam & Goes, 2011; Harris et al., 2020), manta ray site use and movements between Komodo NP sites are likely influenced by prey availability. For example, in Mozambique, higher manta ray numbers were seen at cleaning stations when prey was locally abundant, indicating that manta rays preferentially visit cleaning stations on reefs close to where they are feeding (Rohner et al., 2013). In Komodo NP, the seasonal shifts in the prevailing winds, precipitation, and sea temperature, broadly referred to as the north-west and south-east monsoons, coincide with manta ray visitations increasing in the south during the south-east monsoon and in the central region during the north-west monsoon (Dewar et al., 2008; present study). Seasonal-associated shifts in manta ray abundance also occur elsewhere (Anderson, Adam & Goes, 2011; Couturier et al., 2018; Peel et al., 2019; Setyawan et al., 2018), and it appears that Komodo NP is a large enough area to accommodate seasonal manta ray movements in this region of Indonesia.

Fine-scale oceanographic conditions, bathymetry, and tidal currents appear to affect MA and the central sites/CL differently, likely influencing prey availability and seasonal variations in manta ray site use (Peel et al., 2020). The southern site, MA, faces the Indian ocean and adjacent deep basins and is more directly exposed to the south-east monsoon seasonal upwelling (Ningsih, Rakhmaputeri & Harto, 2013). The resulting increase in local productivity at this site coincides with increased manta ray sightings in the south-east monsoon. Further, tidal currents and tide phase had a lesser effect on visitations in MA (Dewar et al., 2008), presumably because nutrient availability in the south is less dependent on tidal transport. Tidal transport is likely the primary way nutrients and prey are delivered to the central sites (KM and MW), which are sheltered from direct exposure to the Indian Ocean by the large Komodo, Padar, and Rinca islands. In line with this phenomenon, Dewar et al. (2008) found that on the spring tides i.e., during a full and new moon, when currents and tidal transport are highest, manta ray visitations to KM and neighboring sites increased.

Tidal transport of nutrients into the central area would hypothetically occur year-round, allowing for regular foraging opportunities and for manta rays to use the central area year-round. In Germanov et al. (2019b), we noted foraging behavior under specific tidal conditions on the north-west section of KM. However, we recorded fewer feeding events during the south-east monsoon at this location. During the north-west monsoon, when we would expect productivity to be high in the park’s central area, we recorded mass feeding events of up to 30 individuals. Interestingly, ~12 km to the north of KM, the site CL appears to support feeding for some individuals during the south-east monsoon season, when productivity in the north is generally lower. This site is essentially a channel between two islands where tidal current flows, and potentially plankton, are concentrated, providing feeding opportunities for what appears to be primarily a specific demographic of individuals (i.e., immature individuals). Thus, prey availability sustaining optimal foraging conditions for manta rays is likely a factor of seasonal influences on productivity, tidal delivery, and specific topographical features that concentrate prey.

It is vital to continue to investigate local environmental drivers specific to each manta ray aggregation area. Amassing data on these different areas worldwide indicates that environmental predictors for manta ray abundance vary, and unique local factors likely dictate when conditions are most optimal for foraging and cleaning (Barr & Abelson, 2019; Jaine et al., 2012). To predict manta ray site use in other sites within the park and within the region, we could monitor these large-scale shifts in oceanographic conditions with remote sensing of sea surface temperature and surface chlorophyll-a concentration (Dewar et al., 2008; Jaine et al., 2012; Harris et al., 2020; Putra et al., 2020).

Regional connectivity

Long-range movements from Komodo NP to Nusa Penida MPA and elsewhere in the Lesser Sundas (present study; Germanov & Marshall, 2014; Conservation International, 2016), coupled with the lower individual re-sighting rates and site-specific LIRs (i.e., higher degree of transience) in Komodo NP compared to previous reports elsewhere in Indonesia (Germanov et al., 2019a; Perryman et al., 2019), indicate a degree of dispersal from the park’s studied aggregations. However, we did not extensively survey some large areas of the Komodo NP, such as the north of Komodo Island, Padar Island, and the east of Rinca Island, including the islands of Gili Motang and the previously proposed park extension, Gili Banta, where other manta ray aggregations have been observed (E. Germanov, 2017, personal observation). Future research should be expanded to include monitoring in these locations as well as further afield to Sumbawa and Lombok Islands, east of Komodo NP, where satellite telemetry studies and public sightings indicate manta rays visit (Conservation International, 2016; Germanov & Marshall, 2014; MantaMatcher.org).

Links between different populations enable the exchange of genetic diversity (Bonfil et al. 2005; Skomal et al. 2009), bolstering the resilience of populations and ecosystems (Oliver et al., 2015; Sgrò, Lowe & Hoffmann, 2011). These processes are likely occurring with adjacent proposed manta ray subpopulations in the Raja Ampat MPA Network, which show limited exchange (Setyawan et al., 2020). Interchange of Komodo’s manta rays might also be occurring with aggregations in locations where there were historic manta fisheries (Lewis et al., 2015) or elsewhere where studies are not yet completed at adjacent islands in the archipelago, such as Sumba and Rote. Connectivity of the manta rays in the Lesser Sundas region with other Indonesian aggregations, such as those at Raja Ampat, West Papua, and Sangalaki, East Kalimantan is less likely, as they are more distant than the currently known movement range for M. alfredi (1,150 km; Armstrong et al., 2019). Deep ocean basins can be barriers to movement, as has been noted elsewhere (Deakos, Baker & Bejder, 2011; Peel et al., 2020), suggesting that the deep basins between the Lesser Sundas and Raja Ampat, could subdivide the regional population here. In contrast, the interchange along the shallow continental shelf with northern Australia might be more likely (Armstrong et al., 2020). Thus, it is vital to mitigate threats to manta rays outside of the Komodo NP, particularly in potential movement corridors. Tracking studies would provide more information on movement corridors throughout the region, connectivity between conservation zones, and highlight any seasonal trends in long-range movements. Genetic analyses can be used in tandem with telemetry to provide information on population connectivity throughout Indonesia and neighboring countries like Australia, where exchange might occur (Lassauce et al., 2022; Venables et al., 2020).

Local threats

Growing tourism

Identifying key habitats and presence/absence trends can improve the tourism experience by increasing the chances of encounters with manta rays (Barr & Abelson, 2019; Dewar et al., 2008; O’Malley, Lee-Brooks & Medd, 2013). However, it is essential to consider that mass gatherings of manta rays may coincide with crucial feeding, cleaning, social or reproductive events, where minimal disturbance to natural behavior is necessary (Armstrong et al., 2016; Germanov et al., 2019a; Stevens, 2016; Weeks et al., 2015). The pre-pandemic rapid increase in marine tourism at manta ray sites in Indonesia (present study; Germanov et al., 2019a; Purwanto et al., 2021), and a greater than five-fold increase in general tourist numbers from 30,000 in 1996 (Pet & Yeager, 2000) to >175,000 in 2018 (Komodo National Park Office, 2018), doubling from 2014 (80,626) to 2018 (171,830), underscores the need for effective tourism management to avoid potential negative impacts on manta rays (i.e., reviewed by Stewart et al., 2018a; Trave et al., 2017; Tyne, Loneragan & Bejder, 2014). Further, overcrowding of tourist vessels, viewers and divers can reduce the quality of the experience and tourist satisfaction (Mustika, Ichsan & Booth, 2020; Ziegler, Dearden & Rollins, 2012). Tourism in the Komodo NP has increased greatly since the Dewar et al. (2008) study, which aimed to improve the understanding of the spatial distribution of manta rays residing in the Komodo NP and help establish a viable manta ray tourism industry. Additional information is now needed to guide management better to regulate the rapidly expanding marine tourism in the park and prevent undue pressure on the manta rays in this critical habitat (e.g. Division of Boating & Ocean Recreation, 2016; Germanov et al., 2019a; Kasmidi & Gunadharma, 2017; Setyawan et al., 2022; Venables et al., 2016). To combat the rising pressure from tourism, in September 2019, the Komodo NP put limitations on the number of boats and divers/snorkelers allowed in the water simultaneously at KM. While regulations on carrying capacity quotas for boats and divers/snorkelers and appropriate marine conduct within the park were socialized with operators, infringements occur that have warranted the issuance of warning letters (Komodo National Park Office, 2020a). However, repeated offenses can result in more severe sanctions including revoking operational permits, barring entry into the Komodo NP, or legal action (Komodo National Park Office, 2020b).

Tourism boats, carrying between 10 and 35 divers/snorkelers per day, regularly visit manta ray sites (Komodo Dive Operators Community, 2017, personal communication). The central area manta ray sites are the most accessible year-round and receive the bulk of visits, which increased by 34% within 5 years. Thus, manta rays with greater site affinity to the central area are at risk of chronic disturbance—especially KM, which is repeatedly used by most identified manta rays. To cap diver interactions with manta rays, starting September 2019, the number of boats and people allowed to visit KM is 32, each carrying a maximum of ten divers (Komodo National Park, 2019, personal communication). However, at the time of writing, the other manta ray aggregation sites remain without restrictions, and it is foreseeable that caps on KM will displace tourism pressure to nearby MW. The cleaning station at MW is especially small and has lower tourism carrying capacity than KM or MA. Divers commonly overcrowded this area during the peak tourist season (July and August; E. Germanov, 2017, personal observation). Further, tourism boats dropping off and picking up divers commonly drive over the shallow reef housing cleaning stations creating noise pollution, which startles cleaning rays (E. Germanov, 2017, personal observation). Excessive boats and divers on the site could substantially reduce the quality or length of cleaning station visits by manta rays, similar to the disturbance of feeding rays by swimmers elsewhere (Gómez-García et al., 2021; Murray et al., 2020; Venables et al., 2016). Cleaning provides manta rays with a vital health service i.e., removing parasites and facilitating wound healing, and its disruption could affect individual fitness (O’Shea, Kingsford & Seymour, 2010). Further, cleaning stations likely serve as hubs for significant behavioral interactions, such as courtship, and thus reduced visitations might have population-wide implications (Perryman et al., 2019; Stevens, 2016; Stewart et al., 2018a). More divers visiting these reefs also increase the risk of habitat destruction by poor diving techniques (Trave et al., 2017). Increased boat traffic in manta ray feeding aggregation areas likely also increases the risk of boat strikes and propeller injuries to manta rays (Carpentier et al., 2019; McGregor et al., 2019; Strike et al., 2022). In the north of Komodo NP, tourism at CL also appears to be on the rise (E. Germanov, 2017, personal observation). This site is generally accessible year-round, and continued use of this site by as many as 10 or more boats at a time (>100 divers/snorkelers), without strict adherence to best practice codes of conduct, will likely have a substantial impact on foraging manta rays (Murray et al., 2020; Venables et al., 2016). The narrow channel at CL also serves as a passage for boats and this general boat traffic further increases the pressure on manta rays in the area.

Based on our observations, we recommend the following strategies to improve the sustainability of manta ray watching activities in Komodo NP: (1) Enumerate daily boat and tourist numbers that visit all popular manta sites identified in this study to better understand the pressure on sites (CL, KM, MW, MA); (2) Estimate carrying capacity for boats and divers/snorkelers for all popular manta ray sites, taking into account, site size, commonly observed manta ray behavior, coral cover, noise pollution, and diver/snorkeler safety; (3) Place restrictions on boat speeds and minimum distances to foraging manta rays, and known cleaning stations (see Kasmidi & Gunadharma, 2017 for an example of detailed guidelines for activities at a manta ray cleaning station in Raja Ampat, Indonesia); (4) Mark underwater zones of no entry for divers to discourage entry into known cleaning stations (e.g. MA and MW); (5) Display a best practice code of conduct for SCUBA diving and snorkeling with manta rays (see Marine Megafauna Foundation, 2018; Manta Trust, 2018 for examples) at Komodo NP entrances and ticketing offices, tour operator land-based offices, and on tour boats, and mandate verbal briefings visitors prior to manta watching activities; (6) Evaluate tourism compliance through random visits by Park rangers to monitor tour and in-water activities; and (7) Enlist the tourism community in data collection that can assist in adaptive management, such as recording the number of boats on each site, the approximate number of manta rays, identifying photographs, sightings of other megafauna, and threats such as coral damage, poor diving practices, illegal fishing, and marine debris through crowdsourced citizen science platforms, such as divethedata.com and MantaMatcher.org. Further, while outside the scope of this study, marine plastic pollution is identified as a threat to manta rays (Germanov et al., 2019b). In a step in the right direction, recent Komodo NP directives aimed at reducing local waste generation from tourism require tour operators to separate waste and curb single-use plastic use by providing water refills and prohibiting plastic utensils and polystyrene food containers (Komodo National Park Office, 2020b).

Impacts from fisheries and natural predators

Fishing activities continue to threaten manta ray populations in Indonesia to some extent (Booth et al., 2020), even if only by sub-lethal injuries (Germanov et al., 2019a; Stewart et al., 2018a). However, fewer individuals were recorded with fishing gear injuries within the Komodo NP than in the neighboring Nusa Penida MPA (5% vs. 14%; Germanov et al., 2019a), the Maldives (9% for M. alfredi and 5% for M. birostris; Strike et al., 2022), Hawaii (10%; Deakos, Baker & Bejder, 2011), or Florida (27%; Pate & Marshall, 2020). A better understanding of manta ray site use within the park and adjacent areas and limiting fisheries to specific gear types in high-use areas and along movement corridors would enhance conservation measures (Graham et al., 2016). While all the current study sites in the Komodo NP are within areas where fishing is prohibited (Komodo National Park Office, 2020a; Pet & Yeager, 2000), and indiscriminate fishing practices such as gillnet, long-line, and blast fisheries are banned within the park boundaries, in practice non-compliance occurs (IUCN World Heritage Outlook, 2020; Komodo National Park Office, 2020a). These persistent illegal activities are a threat to manta rays within the Komodo NP and need to be urgently addressed.

Approximately 3% of documented manta rays in Komodo NP had predatory injuries, substantially less than the 76% of manta rays in Mozambique (Marshall & Bennett, 2010), 33% in Hawaii (Deakos, Baker & Bejder, 2011), 23% in eastern Australia (Couturier et al., 2014), 15% of M. alfredi and 10% of M.birostris in the Maldives (Strike et al., 2022), but similar to the 2.7% in Ningaloo Reef (McGregor et al., 2019). The potential predators in Komodo NP and greater region include gray reef sharks Carcharhinus amblyrhynchos, silvertip sharks C. albimarginatus, Java sharks C. amboinensis, bull sharks C. leucas, oceanic whitetip C. longimanus, dusky sharks C. obscurus, tiger sharks Galeocerdo cuvier, great hammerhead Sphyrna mokarran, and bluntnose sixgill Hexanchus griseus (White et al., 2006), similar to those proposed for manta rays in Mozambique (Marshall & Bennett, 2010). Of those mentioned above, divers regularly encounter only gray reef sharks and, less commonly, silvertip sharks. Thus, manta rays in this region might be less exposed to predators than in other locations across their range, likely largely attributable to years of intensive shark fisheries in the region (Dharmadi, Fahmi & Satria, 2015; Pet & Yeager, 2000). However, it should be noted that anthropogenic and predator injuries observe and count only those individuals that have survived the encounters and do not provide a full indication of the number of individuals impacted. Modeled mortality rates were very low (~0), similar to those of the Nusa Penida MPA, suggesting that mortality due to fisheries or natural predators is low for the manta rays sighted in the Komodo NP. However, these analyses cannot discern between true mortality or permanent immigration.

Does Komodo NP provide effective protection for manta rays?

Despite current ongoing illegal fishing activities (Herin, 2012; IUCN World Heritage Outlook, 2020; Komodo National Park Office, 2020a), the decade of active enforcement of banned fishing activities in Komodo NP beginning in 1996 (Pet & Yeager, 2000), and passive surveillance by tourism operators, likely afforded some protection to manta rays nearly two decades before the declaration of nationwide manta ray protection. It is conceivable that there may now be spillover from aggregations (Arauz et al., 2019) within the park that could repopulate depleted areas throughout the Lesser Sundas. Until recently, manta rays (mostly reported as M. birostris, although some data predate the Marshall, Compagno & Bennett (2009) species redescription, and few reports of M. alfredi) were actively targeted in fisheries <400 km to the east and west of the Komodo NP (White et al., 2006; Heinrichs et al., 2011; Lewis et al., 2015). These fisheries suggest that additional adjacent manta ray aggregations exist or might become more apparent as populations recover, as observed with other marine species (Pierszalowski et al., 2016; Salton et al., 2021). However, given the manta rays’ conservative life-history traits, local scale repopulation and spillover could take decades (Stewart et al., 2016).

Population abundance and survival remain to be estimated for the manta population in Komodo NP (Couturier et al., 2014; Marshall, Dudgeon & Bennett, 2011). An enhanced understanding of females’ maturity status, pregnancy, and birthing rates are crucial for the regional manta ray population dynamics (Deakos, 2012; Marshall & Bennett, 2010). Longitudinal abundance estimates would also be useful in gauging the impact of tourism at cleaning stations, which has yet to be empirically assessed. Abundance estimates can then be correlated with the number of boats and divers in the water visiting cleaning stations. Further, studies enabling a better understanding of the environmental influences on prey density and foraging in the region will help inform manta population ecology (Anderson, Adam & Goes, 2011; Armstrong et al., 2016; Barr & Abelson, 2019; Dewar et al., 2008; Jaine et al., 2012; Harris et al., 2020; Putra et al., 2020).

Ongoing long-term monitoring is necessary to ensure management and protection strategies successfully safeguard manta rays in the future. To increase the current understanding of manta ray site use, movements, population demographics, and dynamics within the Komodo NP, future studies should expand the survey area to other locations that tracking studies and anecdotal observations have revealed manta rays occupy e.g., German Flag and Padar; (Dewar et al., 2008); East Sumbawa, West Komodo and East Rinca (Conservation International, 2016); Gili Banta (E. Germanov, 2017, personal observation). Studies at additional sites will require a dedicated research program as they are less accessible/less visited by tourism operators, hence are unlikely to be viable for citizen science programs. Surveys in areas likely to contain suitable manta ray habitats have the potential to reveal other aggregation sites important for reproduction and juveniles that should be considered for inclusion into park boundaries. In particular, surveys in the 1990s revealed substantial manta ray aggregations in Gili Banta (M. Erdmann, Conservation International, 2021, personal communication) that contributed to the proposal to expand park boundaries to include the island (Pet & Yeager, 2000). As a priority, future monitoring should be completed to assess whether these aggregations are intact, and, if so, expanding the park’s boundary to include Gili Banta is recommended.

Conclusions

This long-term study confirms that: (1) Komodo NP is an essential habitat for manta rays with several documented high-affinity aggregation sites that individuals show an individual preference for but move within. (2) Surveyed aggregation areas are used differently by adults and immature individuals. (3) A combination of cleaning, reproductive and foraging behaviors occurred at sites demographically dominated by adults, whereas immature individuals were the dominant demographic where foraging was more common than cleaning. (4) Manta ray injuries from fisheries and, to a lesser extent, predation were readily observed, and by-catch in movement corridors outside of MPAs will continue to pressure these depleted populations. Further, tourism use of manta aggregation sites increased during the study and this creates an urgency for developing and implementing a science-based management strategy, monitoring, and adapting the strategy to maintain these essential habitats from undue anthropogenic disturbance. To be sustainable, national conservation strategies for manta rays need to account for the uneven financial benefits that manta tourism provides, as benefits need to reach the communities that still rely on fisheries (Jaiteh et al., 2016; Jaiteh, Loneragan & Warren, 2017; Mustika, Ichsan & Booth, 2020). Nevertheless, it appears that after decades of fishing in surrounding areas, Komodo NP still retains large manta ray aggregations that with careful ongoing management and threat mitigation might allow for regional species recovery in Indonesia. Thus, Komodo NP highlights the benefits of MPAs as a conservation tool for manta rays and underscores the importance of MPAs to be large enough to encompass key aggregation sites and home ranges (Kessel et al., 2017; Peñaherrera-Palma et al., 2020; Setyawan et al., 2020).

Supplemental Information

Supplemental Information 1 Model parameters and fits (ΔQAIC) of manta ray sighting Lagged Identification Rates (LIRs) according to sex and site.

The results for (A) male and (B) female manta rays at Karang Makassar (KM), Mawan (MW), and Manta Alley (MA), Komodo National Park. The records are from January 2013 to April 2018.

Click here for additional data file.

Supplemental Information 2 Model H outputs for time spent within (res time in ± 1 SE) and outside (res time out ± 1 SE) of Karang Makassar (KM), Mawan (MW) and Manta Alley (MA), Komodo National Park.

The records are from January 2013 to April 2018.

Click here for additional data file.

Supplemental Information 3 The combined number of annual survey days at manta ray study sites in Komodo National Park.

Surveys were logged by trained observers (Logs), while ‘Manta Matcher’ surveys also include public presence-only data.

Click here for additional data file.

Supplemental Information 4 The mean individual sightings per dive (n = 617) logged by trained observers.

Figures show mean sightings per (A) year and (B) month in Karang Makassar (KM), Mawan (MW), and Manta Alley (MA), Komodo NP.

Click here for additional data file.

Supplemental Information 5 Manta rays sighted more than once at the Cauldron and their sighting locations.

Colored circles show sighting locations: Cauldron (CL), Karang Makassar (KM), Mawan (MW), and Manta Alley (MA). Solid vertical line indicates the first sighting where the individual is deemed mature. *indicates individuals that were immature throughout the study. The ID codes can be used to call up full sighting records for the individuals from Manta Matcher.org.

Click here for additional data file.

Supplemental Information 6 Individual manta rays sighted in both Komodo National Park and Nusa Penida MPA.

The sighting location (site) and date are plotted for each individual according to their identification (ID) codes. The sex of the individual is indicated within the brackets following the ID code, where (F) indicates female and (M) male. The ID codes, excluding the bracketed sex information, can be used to call up full sighting records for the individuals from ‘Manta Matcher.org’.

Click here for additional data file.

Supplemental Information 7 The proportion of individual manta rays in Komodo National Park plotted by the number of times individuals were sighted.

(A) Combined sightings from the three core sites (Comb.), (B) Karang Makassar (KM), (C) Mawan (MW) and (D) Manta Alley (MA). The total number of individuals for each site and the number of sightings are listed. The records are from January 2013 to April 2018.

Click here for additional data file.

Supplemental Information 8 The mean (± SE) probability of re-identifying an individual manta ray within the same or different central (Karang Makassar and Mawan) core site, Komodo National Park.

The records are from January 2013 to April 2018.

Click here for additional data file.

Supplemental Information 9 Male and female manta ray sighting Lagged Identification Rates (LIR) (± SE) for the three core study areas in Komodo National Park: Karang Makassar (KM), Mawan (MW), and Manta Alley (MA).

The predicted LIRs for models of best fit (Supplementary Table 2) are shown for each group. The records are from January 2013 to April 2018. Standard Errors (SE) are depicted as vertical lines for each data point.

Click here for additional data file.

Supplemental Information 10 Raw manta ray sightings used for modified maximum likelihood residency analyses.

The data details the date, location, and identification code for individual sightings used for the analyses.

Click here for additional data file.

Supplemental Information 11 The raw demographic data used with modified maximum likelihood residency analyses.

The data details the sex and maturity status of each individual identified based on itsidentification code.

Click here for additional data file.

Supplemental Information 12 Raw trained observer logs of manta ray survey days.

Each survey day logged by a trained observer is listed according to the date, location, the time of the start of the survey (time), the duration of the survey (minutes), the number of individual manta rays identified (alfrediID), the number manta rays estimated on the encounter (alfrediEst) and the number of boats present at the site at the time of the survey.

Click here for additional data file.

Supplemental Information 13 Raw manta ray demographic and behavior sighting data.

Demographic and behavioral (beh) data are displayed by date, site and according to the individual’s identification code.

Click here for additional data file.

Supplemental Information 14 Raw manta ray injury data.

Manta rays (individual identification codes) with injuries are listed according to injury type: fishing line, cephalic fin injury (ceph), pectoral fin injury (pect), predatory injuries (pred), in addition to pregnancy status (preg), if applicable.

Click here for additional data file.

This study would not have been possible without the support of Dive Operators Community of Komodo, the local dive community, and many citizen scientists. Notably, we thank, D. Arriaga, J. Arriaga, M. Cobussen, S. Ecob, S. Geier, S. Inderbitzi, D. Keim, N. Longfellow, J. Marlow, F. Nompas, and for their photo contributions, data processing assistance, and overall dedication to Indonesia’s manta rays. Further, we appreciate the data processing assistance provided by L. Auditore, R. Cooper, L. Ellevog, E. Sinderson, and other Marine Megafauna Foundation staff and volunteers. Wildbook provided the use of the online database MantaMatcher.org. We thank G. Winstanley who provided the use of ‘MantaUtil’ for streamlined data processing. We thank H. Whitehead for the guidance he provided on the use of Markov movement models, M. Calver for statistical analysis assistance, D. Chabbane for SOCPROG use assistance, and S. Venables for plotting using R and critical reading of the manuscript. Dharmadi contributed meaningful discussions towards country-wide implications for manta rays in Indonesia. He, unfortunately, passed before the final version of the manuscript was completed, but left a significant legacy for shark and ray conservation in Indonesia in his wake, including his contribution to this study. Bathymetry information used in creating Fig. 1 was available from GEBCO_2014 Grid, version 20150318; www.gebco.net. We thank members of the Faculty of Marine Sciences and Fisheries, Universitas Udayana, Bali, Indonesia for assisting in creating Fig. 1. We are grateful for the assistance of Park Rangers and staff, D. Indrisari, and Y. J. Hamzah. Mark Erdmann and one anonymous reviewer contributed constructive reviews that enhanced our manuscript, especially by providing valuable insights into the historical context of Komodo NP management, shaping the results’ interpretations, and guiding discussions on the recommendations for park management. This article represents HIMB and SOEST contribution numbers 1879 and 11485, respectively.

Additional Information and Declarations

Competing Interests

Author Contributions

Animal Ethics

Field Study Permissions

Data Availability

Ande Kefi is an employee of the Komodo National Park. All other authors declare that they have no competing interests.

Elitza S. Germanov conceived and designed the experiments, performed the experiments, analyzed the data, prepared figures and/or tables, authored or reviewed drafts of the paper, and approved the final draft.

Simon J. Pierce conceived and designed the experiments, analyzed the data, authored or reviewed drafts of the paper, and approved the final draft.

Andrea D. Marshall conceived and designed the experiments, authored or reviewed drafts of the paper, and approved the final draft.

I. Gede Hendrawan analyzed the data, authored or reviewed drafts of the paper, provided logistical support and local knowledge enabling the study, and approved the final draft.

Ande Kefi analyzed the data, authored or reviewed drafts of the paper, provided logistical support and local knowledge enabling the study, and approved the final draft.

Lars Bejder conceived and designed the experiments, analyzed the data, authored or reviewed drafts of the paper, and approved the final draft.

Neil Loneragan conceived and designed the experiments, authored or reviewed drafts of the paper, and approved the final draft.

The following information was supplied relating to ethical approvals (i.e., approving body and any reference numbers):

This study was carried out in accordance with the approval of the Animal Ethics Committee, Murdoch University (R2781/15).

The following information was supplied relating to field study approvals (i.e., approving body and any reference numbers):

This study was conducted under permits issued by the Indonesian Ministry of Research and Technology (Permit #458/SIP/FRP/E5/Dit. KI/XII/2015; Permit Extension#11/TKPIPA/E5/Dit. KI/XI/2016 and #86/EXT/SIP/FRP/E5/Dit.KI/XI/2017) and the Komodo National Park (#SI.1432/BTNK-1/2016 and 2017).

The following information was supplied regarding data availability:

The raw data used for the modified maximum likelihood approach analyses are available in the Supplemental Files.

Additional data on individual manta rays can be obtained by searching the ID number into the publicly accessible database: https://mantamatcher.org/.

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
