# Peer review of "Residency, movement patterns, behavior and demographics of reef manta rays in Komodo National Park"

_PeerJ, doi:10.7717/peerj.13302_

## Round 0.1 · original submission · Major Revisions

I have two very thorough reviews from experts on manta rays in Indonesia. Both recommend focusing the paper on its core findings and not overstating them. This will require major revision from the title to the text. If you the authors are happy to do this please submit a revised version as soon as you can.

·

Excellent Review

This review has been rated excellent by staff (in the top 15% of reviews)
EDITOR COMMENT
This was a most thorough, knowledgeable and constructive review that will improve the paper.

Basic reporting

Overall the article is well-written and easy to follow. I have inserted a number of comments as “sticky notes” directly into the attached PDF with suggestions for small improvements/corrections to the writing, but overall very well-written.

In terms of the literature references and the field background provided, the authors have done a good job with the manta literature and in explaining their approach and methodology. However, the references are woefully incomplete with respect to the conservation background of Komodo National Park. I provide more detailed points on this in the general comments below.

The figures created are clear and well-done, and the raw data is supplied – with the manta sightings data publicly available on the Manta Matcher website.

I do note that the Animal Ethics Approval letter provided here does not seem to be relevant to the present study; it was granted for the purposes of biopsy sampling of manta rays and other filter feeders for a microplastics study. It is not clear to me that the present study would actually require an Animal Ethics Approval letter as it was purely observational and non-invasive, but I simply wonder why a microplastics biopsy sampling Animal Ethics letter is included for this study?

Experimental design

The research presented is within the scope of the journal, and the goals of the study were well-defined, relevant and meaningful. However, the fourth stated goal of the research (an assessment of risks to reef manta rays in Komodo NP to identify management needs) was incomplete in my opinion (see more detailed comments below).

The methods were generally appropriate and well-described, and the “core research” (describing residency, movement patterns and population structure of KNP mantas) was indeed rigorous. I did not feel this was the case with the assessment of the effectiveness of KNP in safeguarding manta rays – as described in significant more detail in my general comments.

Validity of the findings

All underlying data do indeed appear to have been provided, and they seem robust and statistically sound.

As for the conclusions, those related to the core focus of the paper (examining residency, movement patterns and population structure of KNP mantas) are well-stated and interesting. However, I feel strongly that the authors have significantly overstated the conclusions they present with respect to the effectiveness of KNP in safeguarding mantas; the results of their analyses simply do not support this (indeed those results are largely unlinked to conservation success). I provide significant additional commentary in this regard below.

Additional comments

Some of the below comments are also included as “sticky notes” in the PDF, but I’ve pulled out those which I think are most important to my overall assessment and put them below:

-the title of the paper seems inappropriate to me and not a main conclusion justified by the analyses in the paper – which are largely focused on analyzing movement patterns and habitat use and residency/site affinity in Komodo National Park. The title should be reworded to be an “honest advertisement” of the analyses and conclusions justified from these analyses within the paper – something more along the lines of “Residency, movement patterns and population structure of reef manta rays studied at four sites in Komodo National Park”. Also, I would note that Komodo would not qualify as a “large MPA” under most definitions. Internationally, “large scale MPAs” (LSMPAs) are defined as those MPAs larger than 150,000 km2. The authors note the Komodo National Park is only 1817 km2 including land and sea (I would add that it actually has extensions proposed in 1999 which have not yet been enacted that would bring it to 2,321 km2) – but this is still two orders of magnitude smaller than a LSMPA. Komodo NP is moreover an order of magnitude smaller than the “relatively large” MPAs of Indonesia such as Wakatobi or Cendrawasih Bay or Savu Sea marine national parks.

-Lines 46-47: The authors state that characterization of existing threats to manta rays within the park is limited. While this may be true, the authors have not established this sufficiently to make this claim. They do not even cite the 25 year management plan for KNP, nor any of the MANY MPA management reports and surveys produced by The Nature Conservancy (or WWF-Indonesia) in collaboration with PHKA. The primary document for threat assessment in an Indonesian MPA is its 25-year management plan - so without having read and cited this, the authors cannot claim that threats have not been appropriately characterized or addressed.

-Lines 56-57: Mantas from KM and MW, only 5km apart, showed affinity to one or the other site. This is a very interesting finding and I commend the authors on their rigorous data collection to be able to show this – in my mind one of the most interesting specific findings in the paper. However, this further underscores my point about not being able to claim that this is a "population-level" study - well-known manta aggregation sites in Gili Banta, West Komodo, Southwest Rinca and East Rinca and even Padar are anywhere from 20-50km away from the sites studied here - and very likely also are "home" to mantas that show high affinity to those sites. Without having collected data from these other sites, the authors cannot reasonably claim to have conducted a "population level" analysis for KNP mantas.

-Lines 64-67: the authors state that tourism increased over the course of the study – a statement they back up with their own counts of dive boats at their study sites. However, particularly given that the research team was not based on site at all times and only have sporadic data on boat use from each site (and only from a short period of time each day the data was actually collected and might therefore easily have missed boats arriving and diving there earlier or later in the day), and their observations are only of 4 sites, it would seem prudent to examine the actual tourism numbers to the park, which are readily available from the KNP management authority, who collects an entrance fee and records annual visitation numbers. Given that an official from the KNP authority is listed as one of the coauthors on the paper, it is difficult to understand why the authors would not have tapped this important and readily-available source of data for solidifying their finding of increasing tourism numbers. I suggest this should be included in the revised manuscript.

Lines 68-69: Two important notes: 1) the term "zone" has a very specific meaning in the context of Indonesian National Parks - all of which have a zonation plan. In the case of Komodo NP, there are 9 different zone types, none of which are named "marine conservation zone". The authors should refrain from using the term zone with respect to KNP unless it is used in the narrowly-defined legal meaning of the word. 2) I do not find that the authors' results support this claim of KNP safeguarding mantas. Their analysis does not include any estimate of population growth/decline/stability, and particularly given their implied claim that KNP has been protecting manta rays since its inception, they have not included even any qualitative discussion of the conservation of manta rays in KNP for the 30 years' preceding their study.

-Lines 98-99 (also related to the statements on KNP in the Background section of the Abstract): Several important notes: 1) KNP was established in 1980, but the marine extension to the national park was only added in 1984. This needs to be clarified. 2) As noted above, the authors should refrain from using the term "zone" when referring to KNP unless in the legally correct sense of the word. It would be more appropriate to say "..formally established in 1980, primarily for the conservation and management of the endemic Komodo dragon...". 3) Most importantly, the authors claim that KNP is the “first refuge for M. alfredi” and “has functioned as the longest running protected area for manta rays worldwide.” These statements are patently incorrect, and quite surprising given many of the authors have lived or worked in Australia and would be very much aware that the Great Barrier Reef Marine Park was established in 1975 and most definitely is every bit the manta ray refuge that KNP is. Other important (and mostly well-known) MPAs worldwide that protect manta rays and have been established since before Komodo’s marine extension was gazetted in 1984 include: the Galapagos National Park was established in 1959 and began operations in 1968; the Watamu Marine National Park in Kenya was established in 1968 and the inventory done by Cowburn et al 2018 (Atoll Research Bulletin) reports M. alfredi as a key attraction; Bonaire National Marine Park was established in 1979 and is well-known for manta rays; the Poor Knights Marine Reserve in NZ was established in 1981 and has frequent sightings of M. birostris, though perhaps less resident than manta rays in the other MPAs mentioned here; and finally, Kepulauan Seribu is actually Indonesia’s first marine national park (established in 1982) – mantas are now rare and possibly extirpated there, but it would have served as Indonesia’s “first refuge” for manta rays, if simply being gazetted an MPA equates to serving as a refuge (as seems to be implied here).

-Line 124: I do not feel it is appropriate for the authors to characterize this as a "population-level" study when they only investigate four aggregation sites within KNP (and only really significantly analyze data from 3), and do not even include all of the known aggregation sites studied by Dewar et al 2008, nor the many other known manta sites in and immediately adjacent to KNP. The authors do acknowledge this limitation of their study in the final subsection of the Discussion, but they need to tone down the claims of this being a population-level study in the abstract and introduction. This is a very focused study on 4 aggregation sites, not a population-level study.

-Line 125: As mentioned above, it is intriguing to me that the authors purport to assess potential threats to mantas to identify management needs within the KNP, but they do not cite the 25 year management plan for the KNP nor any of the extensive reports and recommendations for improving the management of the marine resources of KNP made by The Nature Conservancy, WWF-Indonesia and other institutions who have worked with PHKA/KSDAE on management of the park since 1995. It may very well be that there is still a need to assess threats to manta rays and identify management needs, but this clearly would need to start with a thorough read (and citation) of the management plan and the many reports that have been produced for the park. In fact, there used to be a rather comprehensive list of threats (as well as an extensive library of the reports on the MPA) on the official park website www.komodonationalpark.org , but that website is no longer functional – another indication of the declining management effectiveness of the park which has been discussed extensively in the press (and which is at odds with the authors’ claims of the effectiveness of the MPA). A close read of the “threat assessment” conducted by the authors in this paper (their study goal #4) shows they looked closely at threats from tourism (at least in the 4 sites they intensively monitored) and conducted their standardized manta photo ID protocols, which includes noting injuries, fishing gear entanglement and attempted predations (bite marks) for each manta photographed. This is of course invaluable data and very much worthy of reporting, but this does not constitute a comprehensive threat assessment for mantas in KNP – which would most definitely need to explicitly address data from park patrols and dive operator reports on incidences of blast fishing, drift gillnets, possible targeted harpoon fisheries, water quality issues including increasing pollution and run-off from the rampant coastal development around Labuan Bajo and the large number of boats likely dumping sewage and ballast water in the park, microplastics (which the lead author is an expert on), etc. Moreover, the authors suggest they are assessing these threats to help identify specific management needs in KNP – but the discussion does not include a concise and clear list of manta management/conservation recommendations to the park authority. The section of the discussion on threats from tourism does contain a number of important observations and recommendations, but many of them stop short of providing easily actionable recommendations to the park authority – eg, “Consideration should be given as to how these codes of best practice could be implemented effectively in KNP.” The authors are experts in manta rays, and if their aim with this paper is to provide manta conservation recommendations to the park authority, they should make a clear subsection in the Discussion section with tight, actionable recommendations to the park authority. If “consideration should be given” as to how codes of practice can be implemented in KNP, the authors should indeed give this consideration and make recommendations based upon their expertise.

-Lines 494-502: The authors note that more immature male rays and fewer mature female rays were observed at CL, and that this site might possibly serve as a nursery for reef mantas. They then list the 3 criteria of Heupel et al 2007/2019 that define an elasmobranch nursery, but do not systematically examine how CL conforms to these criteria. It would seem to me that the authors have a significant enough data set to be able to systematically “test” each of these criteria for the CL site, in much the same way that Setyawan et al 2020 did for each of the sites they proposed as nursery areas for reef mantas in Raja Ampat (and as Germanov et al 2019 did when proposing “Manta Bay” in Nusa Penida as a reef manta nursery). Given the high conservation importance of identifying manta nurseries and providing them extra protections (including from tourism disturbance), I would recommend that the authors make the effort to apply their data to the Heupel et al definition – as identifying Komodo’s manta nurseries and providing recommendations to the park management authority on extending special protections to these areas would be one of the most valuable things a study like this could do from a conservation perspective. I would moreover note that based upon what we do know of reef manta nurseries, and their tendency to be in relatively shallow, protected bays and lagoons, the CL site generally conforms to these physical characteristics, but in fact the large islands of Komodo, Rinca and Gili Banta have NUMEROUS shallow, well-protected bays that might possibly function as manta ray nurseries – something the authors might consider for follow-up study. Of course, that would require chartering a vessel to actively explore the lesser known bays of KNP, but that’s how good conservation science proceeds…

-Lines 506-509: For the discussion of demographic parameters, it would seem appropriate that the authors generally compare their demographic findings not only to Mozambique, Japan and the Maldives, but also to the nearest large population for which such parameters have been calculated - Raja Ampat. Setyawan et al 2020 provide a wide range of observations from an even larger study - and comparing the KNP findings to other populations within Indonesia is clearly appropriate.

-Lines 518-522: I don't follow the logic of this sentence - it is not clear why observations at other manta ray feeding grounds would suggest that immature individuals would disproportionately use German Flag?? The authors seem to be suggesting that immature individuals are more likely to be seen at feeding grounds than at cleaning stations, but they need to reword this particular sentence to be more clear. Moreover, while its good that they recognize the limitation of their study in not actually focusing on the site (German Flag) that had the most recorded manta visitations of any site in the Dewar et al 2008 study, they then seem to imply that German Flag is only a feeding site. German Flag most definitely has several associated cleaning stations! Moreover, our extensive observations of immature reef mantas in nursery areas in Raja Ampat definitely do not support the supposition that immature mantas do not need to clean; each of the nurseries we've examined has numerous associated cleaning stations (which as the authors rightly note, are also important for social interactions, including for juveniles). I might suggest that the reason the authors have not found as many immature individuals in KNP as they have in Nusa Penida is simply because they have not sufficiently explored the many shallow protected bays around Komodo, Rinca, and Gili Banta that are likely preferred nursery habitats.

-Lines 675-678: the authors claim, with no supporting evidence or references, that KNP has likely acted as a refuge for reef mantas 30 years before nationwide manta ray protection. This is most definitely a bold overstatement for which the authors provide no evidence. If they had carefully read the extensive grey literature which exists on KNP marine resource management, they would know that there was very little enforcement of the MPA prior to 1996 when The Nature Conservancy began working closely with PHKA to support management of the MPA (and purchased several patrol boats). Prior to that time, the region was considered a “Wild West” with extensive blast fishing, cyanide fishing, shark finning and large net fishing – all of which would have threatened manta rays. It took a number of years to reign in the illegal fishing, and sadly, once The Nature Conservancy was forced to leave the park in 2010, illegal fishing once again became rampant. This is even reported in the official “Wikipedia” page on Komodo National Park, which states clearly: “After that, more illegal fishermen arrived as enforcement declined greatly following the exit of TNC which had helped fight destructive fishing practices. In early 2012, dive operators and conservationists found many desolate coral sites, reminiscent of grey moonscapes. Illegal fishermen continue to blast sites with 'bombs' in a process known as blast fishing. The fisherman use a mixture of fertilizer and kerosene in beer bottles as explosives, or use squeeze bottles to squirt cyanide into the coral in order to stun and capture fish. In the past two years more than 60 illegal fishermen have been arrested. One of the suspects was shot and killed after he tried to evade capture by throwing fish bombs at the rangers.[14]”
This statement is simply not supported by evidence and should not be made in this paper; it moreover is not related to the methods and analyses of this paper.

-Line 785-787: This statement is rather mind-boggling in its implications; if the authors believe that it is logistically prohibitive to enforce the no-take zones and regulations of KNP, how can they possibly claim that the park has acted as a refuge for 30 years? Paper parks are not refuges. More to the point, however, KNP is a relatively small and readily accessible MPA which at one point in its history (from about 1998-2010) was extremely well-enforced. In comparison to most other MPAs in eastern Indonesia, KNP could hardly be considered to be logistically prohibitive to enforce! The sites discussed in this paper are only 30-50km from the nearby large town of Labuan Bajo, and generally within 5-10km max of ranger stations within KNP. By comparison, 5 of the 9 MPAs in Raja Ampat are 170-210 km from the nearest large town of Sorong. Those MPAs ARE logistically challenging, yet they have regular patrols that have largely eliminated illegal and destructive fishing. This statement should be removed - it is not in line with conservation reality in Indonesia and is moreover at extreme odds with the claims of the paper about the conservation success of KNP.

-Line 817: This statement suggests that the authors primarily engage with only one component of the tourism sector in KNP - the land-based operators located in Labuan Bajo. I have made perhaps 30 liveaboard dive vessel trips to Komodo since 1996, and I have visited German Flag and Padar Kecil on most of those trips. The longer-term liveaboard operators, many of whom have extensively explored KNP over the past 25 years, and who are catering to a much more experienced diver clientele than the operators based out of Labuan Bajo, actually largely avoid the sites the authors have mentioned and studied in the north of the park due to the overcrowding of day boats there. If the authors are serious about truly surveying KNPs mantas at the "population level", engagement with these liveboard operators that travel much further afield in KNP (including to Gili Banta and Gili Motang and West Komodo) would be a clear next step for their efforts.

-Line 840: While I would agree that it is almost certain that KNP has provided SOME protection to reef manta rays (interestingly, this protection is likely most directly attributable to the de facto enforcement role which the tourism boats play at the main manta aggregation sites), without any baseline data to compare to, nor data on population growth, nor any historical data or observations reported in this study, it is only appropriate to suggest this as a hypothesis/possibility and certainly not a main conclusion of the paper (and consequently should not be part of the title of the paper).

-Line 845: Normally, in a conservation-focused paper, the authors might conclude with a set of recommendations explicitly targeted to the management authorities in charge of managing an MPA or other jurisdictional waters. The recommendation to expand the survey area, which the authors recognize will require a dedicated research program, is intriguing in this regard. To whom is this recommendation made? At the present time, it seems the only ones conducting these manta surveys in KNP are the authors themselves, in which case are they making a recommendation to themselves? If this is the case, it would seem the more appropriate way to state this would be “To overall increase the current understanding of site use , movements…., we hope to be able to expand our surveys to other locations….”. Or “we intend to”. If it is a recommendation, the authors need to clarify to whom they are recommending this.

In summary, the authors have done an excellent job of analyzing an important data set and providing some very interesting insights on residency, movement patterns and population structure of reef manta rays in Komodo NP. Certainly this dataset and the analyses included are worthy of publication and a useful contribution to the reef manta literature, and I recommend that the article be accepted after some important revisions. I have provided a number of small recommendations to improve the clarity of the narrative, but more importantly I have listed above a number of concerns about statements the authors have made which I believe are either inaccurate (eg, that KNP is the world’s first manta reserve) or significantly overstate what can be concluded from the data and analyses presented here. At its core, this is a solid scientific study with some important findings. However, my two main concerns relate to the limited geographic focus of the study (4, and often only 3, manta ray aggregation sites for which the authors analyze data, despite there being at least three times that number of known aggregation sites within and immediately adjacent to the park) and the unsupported claims about the conservation effectiveness of KNP in safeguarding mantas. My suggestion would be that the authors simply revise the title of their paper to be more in line with their actual analyses, and tone down their claims about the comprehensiveness of this study and the conservation effectiveness of KNP. They should also consider including a tight, concise recommendations section for the park management authority based on their analyses and expert opinion on what needs to be improved in the KNP to better protect manta rays.

If, on the other hand, the authors want to properly examine the conservation effectiveness of KNP for manta rays, it will require a MUCH more detailed study and description of the long and tortuous history of KNP MPA management, including a more thorough description of the conservation setting of the park (including that it is a Man and Biosphere Reserve and a UNESCO World Heritage Site). The authors will need to not only thoroughly familiarize themselves with the KNP 25-year management plan, but also need to dive deep into the grey literature (as there is an ENORMOUS amount of project reports, surveys and management recommendation papers that were produced in the period 1995-2010 for KNP) and interview the conservation professionals who designed KNP’s management plan, as well as those that were involved in the management after 2010 when The Nature Conservancy left KNP. They also would need to interview many of the long-term tourism operators that were exploring KNP in the 1990s and early 2000s, and provide historical context and observations on KNP’s manta population – this data absolutely DOES exist, but it is not published in the peer-reviewed literature and requires a much more intensive approach to data collection. Unfortunately, this is typical of the conservation histories of almost all MPAs in Indonesia and much of the developing tropics – publication in peer-reviewed literature is not a priority for conservation organizations and government agencies working hard to implement and manage MPAs. Finally, the authors would also need to provide some indication of the stability or growth of the KNP manta ray population to be able to claim that KNP is effectively safeguarding its manta rays. They did show that their best-fit resighting model suggested low mortality, but that does not seem sufficiently robust to claim conservation success, particularly in light of the abundant public criticism of the current lack of management effectiveness of the park. At any rate, such an analysis would most definitely be an extremely worthwhile endeavor and a most useful paper – but unfortunately, in my strong opinion the data and analyses presented in the current study are not sufficient to address the question of the relative effectiveness of KNP in safeguarding reef manta rays. This would require a rather major revision to the manuscript.

-My final comment – I noted in one of my comments that I recorded more sites with manta rays during my first two-week survey of Komodo in 1995 than the authors have recorded for the whole of their 14 year survey. This is not meant as a major criticism, but rather a “reality check” on the large swaths of KNP that are not addressed in this paper. I of course should not criticize what I would classify as “unadventurous science”, but likewise the authors should not overclaim the utility and representativeness of their study, when it was in fact confined to a small area of a relatively small park. The authors conclude that further research is needed to identify and protect other critical aggregation areas within the region – indeed, but I cannot stress enough that Komodo is not a very large park and there has been an ENORMOUS amount of marine survey effort put into this region, dating all the way back to Salm and Halim 1984 and including numerous surveys in the nearly 4 decades since by NGOs, the Indonesian Institute of Sciences and various local universities. The authors do not cite any of these papers or reports, and while they most definitely will not include the type of detailed data the authors have been collecting, there is likely a wealth of anecdotal observations on manta rays that would prove useful to any truly comprehensive study on KNP mantas with a focus on putting forth recommendations to improve their conservation and management. As a single concrete example, surveys of the park in the late 1990s revealed significant aggregations of manta rays (as well as some beautiful reefs and large aggregations of reef fish) at Gili Banta, a large island to the NW of Komodo Island. As the 25-year management plan for the park was being prepared, a proposal was also put forth in 1999 to expand the park to include Gili Banta – both to include the important marine resources around the island as well as to prevent Banta from becoming a base for illegal fishers and deer poachers to make excursions into the park. Sadly that does not appear to have been acted upon, but it nonetheless still exists as a formal proposal to the government that could be revived. If the authors are keen to truly improve the conservation of manta rays in Komodo NP, I believe an exploration of the current status of the manta aggregation sites in Banta would be a top priority, and assuming they are still intact (anecdotal evidence from liveaboards pre-Covid pandemic suggests they were), expanding the park’s boundary to include Banta would certainly be a top conservation recommendation.

Reviewer 2 ·

Excellent Review

This review has been rated excellent by staff (in the top 15% of reviews)
EDITOR COMMENT
This was a most thorough, knowledgeable and constructive review that will improve the paper.

Basic reporting

- The language use is this paper is overall clear and unambiguous.
- Literature referenced by the authors were not incomplete to provide a strong background written in Introduction. This also lead to misleading and incorrect overstated claim on the longest running protected area for manta rays worldwide. Another example of insufficient literature referenced is that the authors didn't cite the management plan of Komodo National Park when they intended to discuss the effectiveness of this park to protect manta rays. In the context of manta ray conservation in Komodo and Indonesia in general, the authors insufficiently referenced several important milestones preceeding the full protection of manta rays in Indonesia in 2014, including regional level protection for manta rays in Raja Ampat in 2012 and importantly in Manggarai Barat (Komodo) in 2013.

I recommend the authors to make correction on the incorrect overstated statements and fully acknowledge manta ray conservation efforts in place in the region and Indonesia fairly.

Experimental design

The research questions are well defined and are useful to fill knowledge gap on the absence of population level studies investigating manta ray demographics and habitat use within Komodo National Park. I also commend the authors for strong analyses performed in this study.
Despite this, it is unclear to me whether the sighting data submitted by public/citizen science were also included in the analyses combined with contributed data.

Validity of the findings

In this paper, the authors made an effort to link between the main findings (site affinity and population structure) and conservation and management effectiveness to protect manta ray in the park. However, this was done without referring to management plan of the Komodo National Park. Moreover, instead of using 14 years of data, the authors in fact used limited data from 2013 to early 2018 contributed by citizen science covering four manta aggregation sites in the park. The authors also emphasized the crucial role of the park for regional population recovery. However, the arguments for this statement were invalid as noted in my comments in the manuscript. Apart from this, I note that most part of the discussion on environmental influences on seasonal movements were unnecessary as this paper didn’t include any environmental data in the analyses.

Please see my suggestions in the reviewed manuscript.

Additional comments

Overall, this study provide important findings on the site affinity, population structure, and behaviours of reef manta rays in Komodo National Park. However, there several major points must be addressed in order to make this study more suitable for publication. I strongly suggest that the authors consider focusing only on the main findings, which were supported strong statistical analyses, and then briefly discuss about the management implications of these main findings. I also suggest that the authors should address some more detail corrections in the annotated manuscript.

I also note that the permits (including permit extensions) issued by the Indonesian Ministry of Research and Technology allowing the first author was granted only from 2015-2017. It is unclear to me whether these permits also covered the research in Komodo National Park or in addition to Nusa Penida. It is also important to note that the permit from the Komodo National Park management authority was issued only for 2016, while the sighting data collected from 2013-2018. I am afraid that the data can be used in the analyses only include those collected in 2015-2017 or even only in 2016. Please provide more clarification on this.

Annotated reviews are not available for download in order to protect the identity of reviewers who chose to remain anonymous.

---

## Round 0.2 · Minor Revisions

The referees are very pleased with the revised paper. They make some good suggestions regarding the Abstract, inference of a nursery area may be mistaken if size estimates were incorrect, and essential corrections to incorrect citations of the literature. Please review all literature comparisons to double-check they are accurate as such mistakes will reflect poorly on the authors. These were particularly helpful reviewers and you may wish to acknowledge them.

·

Basic reporting

Overall the article is well-written and easy to follow. I have again inserted a number of small comments as “sticky notes” directly into the attached PDF with suggestions for small improvements/corrections to the writing, but overall very well-written.

In terms of the literature references and the field background provided, the authors have done a good job with the manta literature and in explaining their approach and methodology (I did suggest considering adding one brand new reference – Lassauce et al 2022 – on using genetic methodologies to look at population connectivity).

The figures created are clear and well-done, and the raw data is supplied – with the manta sightings data publicly available on the Manta Matcher website.

Experimental design

The research presented is within the scope of the journal, and the goals of the study were well-defined, relevant and meaningful.

The methods were generally appropriate and well-described, and the “core research” (describing residency, movement patterns and population structure of KNP mantas) was indeed rigorous. The authors moreover did an excellent job of taking on board a number of my previous comments and suggestions related to providing meaningful recommendations to improve conservation and management of manta rays in KNP.

Validity of the findings

All underlying data do indeed appear to have been provided, and they seem robust and statistically sound, with the conclusions well-stated and interesting.

Additional comments

Overall the authors have done an admirable job of significantly “tightening” this manuscript, focusing in on detailed analyses of the valuable dataset they’ve collated and providing a number of valuable insights on reef manta ray residency, movement patterns, behavior patterns and demographics in KNP. I was particularly pleased to see their analysis of tourism growth and impacts on manta rays in the park and the effort put into providing clear, actionable recommendations to the KNP Management Authority (and tourism operators) to improve management of manta ray tourism, and in outlining future research priorities in KNP and the immediate surrounds to best inform conservation and management.

Other than addressing the various minor comments and corrections I’ve provided, my only overarching suggestion is for the authors to reconsider their abstract to ensure it conveys all of the points the authors consider most important from this study. There’s a LOT covered in the study, and the abstract is necessarily short and can only cover key points – but I note that for instance none of the discussion of tourism impacts nor the recommendations included in the discussion section are mentioned in the abstract. That of course is the prerogative of the authors to choose what they most want to highlight – I would simply note that I think there are some key findings/recommendations that at least I found very compelling that are not in the abstract.

With minor revision, this paper is ready for publication and will be a valuable contribution to our knowledge of reef manta rays while also providing excellent recommendations to the KNP Mgmt Authority and laying out a clear vision for future manta ray research priorities in KNP.

Reviewer 2 ·

Basic reporting

- The language use is this paper is overall clear and unambiguous.

- Figure 1 still need a minor improvement. As I suggested previously, the label of Raja Ampat is still misplaced. I suggest using a dot and line to show exactly where it is. Need to add unit for bathymetry (in the legend).

- Good to see that the authors have made significant improvement in the literature references. However, in some references, the authors made incorrect citations by cherry-picking findings/statements from the cited papers to support findings in the manuscript. For example, in line 574, the authors cited partially some findings on the number of immature individuals from Raja Ampat.

Setyawan et al 2020 (page 62) stated “In total, 153 juvenile M. alfredi (individuals ≤ 2.4 m DW) were documented in Raja Ampat between 2011 and 2019….” and this statement followed by another statement “Focusing on YoY individuals (≤ 2 m DW), we recorded a total of 65 YoY individuals between 2011 and 2019…”.

Here the authors made incorrect comparison as they compared immature individuals in Komodo and Nusa Penida with only YOY individuals, but not including other juveniles in Raja Ampat. Moreover, Setyawan et al 2020 focused on the number of juveniles and YoYs sized ≤ 2.4 m DW), and didn’t summarise the number of ALL immature individuals >2.4 m. Please note that males were estimated to reach maturity at 270-280 cm in DW (Stevens 2016).

Another example for incorrect statement can be found in line 43-45, where the authors stated "In 2013, manta ray fishing was banned in the waters within and adjacent to Komodo NP, paving the way for a nationwide manta ray fishing ban in 2014.".

Setyawan et al 2022 wrote in details about the manta conservation in Indonesia. While the authors wrote some statements about manta conservation milestones in line 115-120, the statement in the Abstract (lines 43-45 was incomplete and misleading).

I strongly recommend the authors to fix those incorrect and incomplete citation and misleading statements. They authors have a significant effort to fix many overstated claims in the the previous version of the manuscript, and therefore similar effort should be taken to fix those in the final manuscript.

Experimental design

The research questions are well defined and are useful to fill knowledge gap on the absence of population level studies investigating manta ray demographics and habitat use within Komodo National Park. I also commend the authors for strong analyses performed in this study. However, there is an important issue regarding the methods described, in particular size estimation that later in the discussion the authors made a conclusion about the potential nursery in CL.

While the authors described in details about manta identification and maturity status in lines 195-208), the authors didn’t add any more information about size estimation, although they mentioned in line 189 that size estimation was also collected by the approximately 20 local dive operator staff as stated in line 186.

In this study, I wonder if the local dive operator staff did collect the size estimation data while dealing with guests, which is their main responsibility I believe? Furthermore, how to maintain accuracy and minimize biases of the size estimation given many local dive operator staff and citizen scientists involved in the data collection. What is the resolution of size estimated?

While the authors clearly admitted the lack of size estimation accuracy in study (line 347), they have to stated in details on how size estimation was done. The authors didn’t report the numbers of individuals with size estimated as well.

I strongly recommend adding the details of size estimation as they did for manta identification and defining maturity.

Validity of the findings

The authors made an improvement in the validity of the findings in this manuscript. However, the authors (in lines 588-590) made invalid conclusion about the possibility of CL as a nursery area using criteria proposed by Heupel at al 2007 on nursery area for newborn and young-of-the-year/YoYs (in this manuscript, the authors use yearlings to represent those newborn and YoY individuals.

With insufficient survey effort at CL and biases and inaccuracy in size estimation (if there were size estimations done), the authors do not have any qualification to test the criterias proposed by Heupel et al 2007 to conclude that CL is a potential nursery area of manta rays.

The authors noted (in line 585-587) “….we did not have the data to be able to discern yearlings from generally immature individuals, an important distinction when considering nursery habitats”, this is a clear message that the authors are not able test the criteria. It is unclear weather the immature individuals encountered in CL were yearlings or other immature individuals which sized were up to 270 cm for males (Stevens 2016). IF the immature individuals were in fact yearlings, the authors didn’t have data to fulfil criterion (2), which is the yearlings remain in the area for extended periods, as they didn’t have continuous monitoring effort in CL, but only opportunistic effort.

Given these, there is no point of testing the criteria for nursery in this manuscript. While identying manta nursery area is critical for conservation (Stewart et al 2018), much more significant and dedicated effort is required. Relying heavily on citizen science and inconsistent survey effort (survey period and data collector) are not sufficient, even for generating too early conclusion on potential nursery area.

Finally, in this manuscript, I strongly recommend the removal of discussion about CL as potential nursery area as well as the assertive by too early effort to test the criteria for nursery area.

Additional comments

In conclusion, I appreciate the authors that have made significant effort in addressing issues raised in the previous version on the manuscript. However, there are some critical statements and conclusions requiring serious effort from the authors to fix/address for the acceptance of this paper.

Overall, I recommend this article should be accepted after the authors addressing and fixing the following main issues: 1) incorrect/inaccurate statements/citation as described in details above; 2) size estimation - accuracy and biases; and 3) assertive effort to test criteria for nursery area.

Annotated reviews are not available for download in order to protect the identity of reviewers who chose to remain anonymous.

---

## Round 0.3 · accepted · Accept

Thank you for the most thorough and respectfully considered revisions so the paper. thank you for publishing this important work in PeerJ.